# Chromosome arm aneuploidies shape tumour evolution and drug response

Ankit Shukla [1,7], Thu H.M. Nguyen[1,7], Sarat B. Moka [2], Jonathan J. Ellis[1], John P. Grady [1], Harald Oey[1], Alexandre S. Cristino[3], Kum Kum Khanna[4], Dirk P. Kroese [2], Lutz Krause [1], Eloise Dray[5], J. Lynn Fink [1] & Pascal H.G. Duijf [1,6]*

Chromosome arm aneuploidies (CAAs) are pervasive in cancers. However, how they affect cancer development, prognosis and treatment remains largely unknown. Here, we analyse CAA profiles of 23,427 tumours, identifying aspects of tumour evolution including probable orders in which CAAs occur and CAAs predicting tissue-specific metastasis. Both haematological and solid cancers initially gain chromosome arms, while only solid cancers subsequently preferentially lose multiple arms. 72 CAAs and 88 synergistically co-occurring CAA pairs multivariately predict good or poor survival for 58% of 6977 patients, with negligible impact of whole-genome doubling. Additionally, machine learning identifies 31 CAAs that robustly alter response to 56 chemotherapeutic drugs across cell lines representing 17 cancer types. We also uncover 1024 potential synthetic lethal pharmacogenomic interactions. Notably, in predicting drug response, CAAs substantially outperform mutations and focal deletions/amplifications combined. Thus, CAAs predict cancer prognosis, shape tumour evolution, metastasis and drug response, and may advance precision oncology.

[1] University of Queensland Diamantina Institute, The University of Queensland, Translational Research Institute, 37 Kent Street, Brisbane, QLD 4102, Australia. [2] School of Mathematics and Physics, The University of Queensland, Brisbane, QLD 4072, Australia. [3] Griffith Institute for Drug Discovery, Griffith University, 46 Don Young Rd, Nathan, QLD 4111, Australia. [4] QIMR Berghofer Medical Research Institute, 300 Herston Road, Herston, QLD 4006, Australia. [5] Department of Biochemistry and Structural Biology, UT Health San Antonio, 7703 Floyd Curl Drive, San Antonio, TX 78229-3900, USA. [6] Institute of Health and Biomedical Innovation, School of Biomedical Sciences, Faculty of Health, Queensland University of Technology, 37 Kent Street, Brisbane, QLD 4102, Australia. [7] These authors contributed equally: Ankit Shukla, Thu H.M. Nguyen. *email: pascal.duijf@qut.edu.au

Cancer cells typically adopt a number of features that distinguish them from normal cells, such as the ability to proliferate in an uncontrolled manner[1]. The acquisition of such malignant properties invariably has a genetic or genomic basis[2,3]. For instance, oncogenes may harbour activating mutations or be subject to amplification, while tumour suppressor genes may acquire inactivating mutations or suffer from copy number loss. The gain of mutations and copy number changes, collectively referred to as genomic instability, are common in cancer cells and are potent drivers of tumorigenesis, intra-tumour heterogeneity and drug resistance[1–5]. A range of mechanisms of genomic instability have been identified[3,6]. Point mutations can be caused by extrinsic or cell-autonomous factors, including defective DNA replication or DNA repair pathways[7–9]. Somatic copy number changes may emerge following replication stress, telomere crisis, defects in DNA damage response and repair pathways or chromosomal instability (CIN), which result in structural or numerical chromosomal abnormalities[3,6,10].

Cancerous copy number aberrations include small segmental deletions and amplifications, genomic regions of multiple megabases, gain or loss of chromosome arms or whole chromosomes and whole-genome doubling. Many translational studies have focussed on individual chromosomal aberrations in cancer, such as HER2 amplification in breast cancers[11,12], whereas others have studied combinations of copy number changes. For instance, simultaneous loss of chromosome arms 1p and 19q occurs frequently in gliomas and strongly predicts good patient outcome[13].

Other studies determined somatic copy number alteration (SCNA) landscapes across individual or multiple cancer types using systems genomics approaches. For example, tumour cells were found to depend on focal amplifications of anti-apoptotic genes in order to survive[14]. Loci on chromosome arm 9p were identified as potential cancer drivers and therapeutic targets in lower grade glioma[15]. Moreover, aneuploidy and SCNA levels in cancers were shown to positively correlate with mutation load and cell proliferation, while negatively correlating with immune cell infiltration and patient survival in immunotherapy trials[16–18]. Finally, loss of chromosome arm 3p is common in squamous tumours[17,19,20].

A recent landmark study identified a large number of mutations and focal SCNA-drug interactions across many cancer types[21]. Functionally, such associations between focal SCNAs and drug response may be easily understood, if the drug target is encoded by a gene located on the focal SCNA. However, it is possible that more complex pharmacogenomic interactions exist. Compared to those of focal SCNAs, the frequencies of chromosome arm-level aneuploidies (CAAs) are about 30 times higher than expected based on the inverse-length distribution of focal SCNAs and this phenomenon is widespread among cancer types[14]. In addition, CAAs on average affect about 25% of the cancer genome, whereas focal SCNAs involve 10%[14]. Thus, CAAs may profoundly affect how cancer cells respond to drug treatment. Yet, this has not been thoroughly investigated.

Here, we determine the frequencies of CAAs in 31 cancer types. In-depth CAA analyses provide insights into tumour evolution. In addition, we identify 160 individual CAAs or co-occurring CAA pairs that predict good or poor cancer patient prognosis. Finally, using machine learning, we identify CAAs that predict increased sensitivity or resistance to dozens of chemotherapeutic drugs and show that CAAs are considerably stronger predictors of drug response than mutations and focal SCNAs combined.

## Results

**Pan-cancer chromosome arm aneuploidies**. Using Genome-wide SNP6 Array data from The Cancer Genome Atlas (TCGA),

we determined numerical CAAs in 11,019 human tumour samples across 31 cancer types (Supplementary Data 1; see Methods). We used the largest dataset (breast cancer, BRCA) to directly compare our CAA frequencies to CAA frequencies determined: (1) by Taylor et al.[17], who also used TCGA SNP6 array data, (2) using TCGA whole-genome array copy number data, (3) using SNP6 array data from an independent dataset, METABRIC[22] and (4) using ICGC/PCAWG (Pan-cancer Analysis of Whole Genomes) whole-genome sequencing data[23]. These comparisons showed strong correlations with Pearson coefficients of $r = 0.9280$, $r = 0.9106$, $r = 0.9688$ and $r = 0.6154$, respectively (Supplementary Fig. 1a–e). Thus, this provided both technical and biological validation.

**Cancer type-specific bias towards chromosome arm gain/loss**. In comparing CAA burden between cancer types, we noticed that haematological cancers accrue significantly fewer CAAs per tumour than solid tumours (median 0 and 5, mean 1.5 and 6.5, respectively; $p = 2.6 \times 10^{-60}$, Mann–Whitney $U$ test; Fig. 1a). However, per cancer type, the average CAA burden ranged considerably, from 0.5 to 14.7 (Fig. 1b). Importantly, we thus far determined CAAs in tumours irrespective of whether they had undergone whole-genome doubling (WGD), a common phenomenon in tumours[24,25]. We found that CAA burden is higher in WGD-positive (WGD+) samples than in WGD-negative (WGD−) samples, however, increased CAA burden in solid cancers compared to haematological malignancies is independent of WGD status (Supplementary Fig. 2a, b).

Per sample, haematological cancers show more gains than losses, whereas solid cancers show more losses than gains ($p = 0.0167$ and $p = 1.77 \times 10^{-36}$, unpaired Mann–Whitney $U$ test) (Fig. 1c). Within samples, this difference is even more significant ($p = 0.0015$ and $p = 1.08 \times 10^{-111}$, paired Wilcoxon signed-rank test) (Fig. 1c). The bias towards loss in solid cancers is independent of WGD status and the bias towards gain in haematological tumours applies to at least the vast majority (158/171 = 92%) of these malignancies that do not undergo WGD (Supplementary Fig. 2a–d).

Thus, we conclude that haematological cancers preferentially gain chromosome arms, whereas solid tumours exhibit a bias towards arm loss, irrespective of WGD status.

**CAA frequencies reveal aspects of tumour evolution**. Subsequent to our global observation of opposite chromosome arm gain/loss biases in haematological and solid cancers (Fig. 1c), we performed several in-depth analyses. We first determined how many individual CAA-positive tumours show more chromosome arm gains than losses ($G > L$), equal numbers of gains and losses ($G = L$) or more arm losses than gains ($G < L$). Consistent with our previous observation, this analysis indicates that haematological cancers show a strong bias towards chromosome arm gain, while solid tumours preferentially lose chromosome arms ($p = 4.0 \times 10^{-5}$ and $p = 5.9 \times 10^{-104}$, Chi-square test) (Fig. 1d).

We next assessed whether these opposing biases were dependent on the total number of CAAs per tumour. For haematological cancers, the fractions of tumours with $G > L$ were always higher than expected by chance (except when #CAAs = 5) (Fig. 1e). However, solid cancers with 1 or 2 CAAs also showed a significant bias towards gain (Fig. 1e). In contrast—and in line with our expectation given observations in Fig. 1c, d—solid cancers with 3 to 17 CAAs per tumour consistently showed a significant bias towards chromosome arm loss (Fig. 1e). Interestingly, for solid cancers, there is a slight shift in the 'turning point' from bias towards gain to bias towards loss depending on WGD status. In WGD− samples, this turning point is between 2

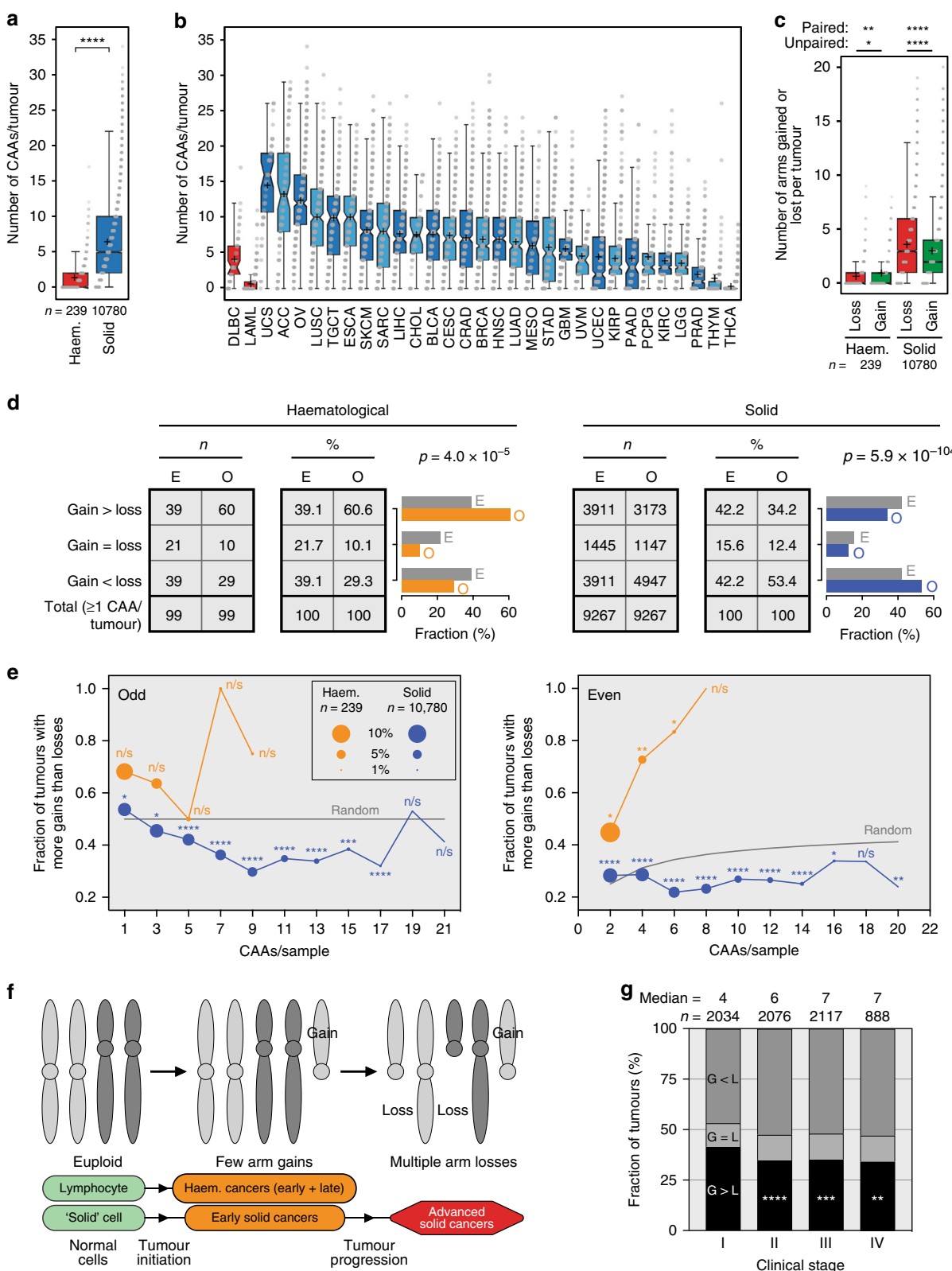

and 3 CAAs/sample, whereas for WGD + samples, this point also roughly doubles, as it is between 4 and 5 CAAs/sample (Supplementary Fig. 3a–d). Collectively, these data suggest that during tumorigenesis, haematological cancers gain few arms, whereas solid cancers initially often also gain few chromosome arms, but at later stages they preferentially lose multiple arms (Fig. 1f).

Consistent with this, solid tumours show increasing CAA burden as they progress to more advanced stages (Supplementary Fig. 4a). In addition, the fraction of CAA-positive tumours with $G > L$ is significantly higher in stage I than in stage II–IV solid tumours (Fig. 1g, Supplementary Fig. 4c). This is typical for individual solid cancer types (Supplementary Fig. 4e) and both WGD− and WGD+ tumours show larger $G > L$ fractions in stage

**Fig. 1 CAA frequencies provide insights into tumour evolution. a** Box plot comparing CAA burden per tumour for haematological and solid cancers. Shown are mean (+), median with 95% confidence intervals (notches), interquartile ranges and all data points. *P* value: Mann–Whitney *U* test. **b** Box plot as in **a** showing CAA burden per cancer type. Abbreviations of each cancer type are shown in Supplementary Data 1. **c** Box plot as in **a** showing the number of chromosome arms lost or gained in haematological and solid cancers. *P* values: Mann–Whitney *U* test (unpaired), Wilcoxon signed-rank test (paired). **d** Contingency tables showing expected (*E*) and observed (*O*) numbers (*n*) and percentages (%) of CAA-positive haematological and solid tumours with indicated arm-level gain:loss ratios. Bar graphs show the respective expected and observed fractions. *P* values: Chi-square tests. **e** Shooting star plots showing fractions of tumours with *G* > *L* as a function of the total number of CAAs per sample. Odd and even numbers are shown separately. Dot sizes are proportional to the fractions of haematological (orange) and solid tumours (blue). *P* values: binomial tests. **f** Tumour evolution model showing that both haematological and solid cancers initially gain few chromosome arms, whereas only solid cancers subsequently preferentially lose chromosome arms. **g** Distributions of CAA-positive solid tumours with indicated intra-tumour chromosome arm gain (*G*):loss (*L*) ratios according to clinical stage. Median CAA burden and sample sizes are shown for each stage. *P* values: $p = 7.1 \times 10^{-5}$, $p = 1.4 \times 10^{-4}$, $p = 0.0013$, respectively, Chi-square tests relative to stage I. *P* value abbreviations are defined in the Methods section. Source data are provided as a Source Data file.

I (Supplementary Fig. 4f, g). We note, however, that even in stage I solid tumours, the median CAA burden is above the CAA burden 'turning point' that marks the switch between bias towards gain and bias towards loss (Fig. 1g, Supplementary Figs. 3a–d, 4f, g). This heterogeneity probably explains why even stage I tumours collectively show a slight bias towards loss.

**CAAs in primary and metastatic tumours.** Other features of tumour evolution may surface from studying metastasis, a key feature of advanced tumours. We used the MSK-IMPACT dataset[26] to compare 5778 primary solid tumour samples to 4424 metastatic solid tumour samples (Supplementary Data 2). Metastatic samples mostly show increased CAA burden compared to type-matched primary samples (Fig. 2a, Supplementary Fig. 5a). Strikingly, irrespective of the primary site, brain metastases show a significantly higher CAA burden than metastases at other common sites (Fig. 2a, Supplementary Fig. 5b) and this is independent of WGD status (Supplementary Fig. 6a–c).

We also identified individual CAAs that associate with metastasis, most significantly for prostate and non-small cell lung cancer (Fig. 2b, Supplementary Figs. 5c, 6d). Notably, these are mostly tumour type-specific; there is no universal pan-cancer CAA linked to metastasis (Fig. 2b, Supplementary Fig. 5c). However, several CAAs show moderate to strong specificity for metastasis specifically to liver, bone or brain (Fig. 2c, Supplementary Fig. 5d), although WGD status affects this specificity for liver metastases (Supplementary Fig. 6e). Interestingly, some CAAs that are acquired by brain metastases are also common in primary brain tumours (Fig. 2c, Supplementary Fig. 6e).

**A stochastic model for CAAs in breast cancer.** We built a tree with *k* levels to model the sequential acquisition of CAAs during tumour evolution (see Methods, Supplementary Fig. 7). On each level *k*, nodes referred to 'CAA karyotypes' and included all possible karyotypes with exactly *k* CAAs. Edges, all directed from level *k* to level *k* + 1, referred to transitions between karyotypes acquiring one additional CAA. Using the frequencies of each observed CAA karyotype in the breast cancer (BRCA) dataset, we estimated the probability of acquiring one specific CAA before another during breast cancer development. This resulted in a 78 × 78 transition probability matrix (Supplementary Data 3). Along with the observed frequencies of each individual CAA, we generated a network for predicted high-probability sequential acquisitions of CAAs (Fig. 2d). Examples of such transitions of various path lengths include: (+19q → +1q), (−12q → −17p → +1q) and (+7p → −7q → +3q → −13q → +8q). Thus, this provided probable orders in which CAAs are acquired during cancer development.

**72 CAAs predict patient survival outcome.** We next performed robust multivariate patient survival analyses using cancer type, clinical stage, age and all univariately significant CAAs as covariates. This identified 36 CAAs significantly associated with overall survival and 36 CAAs associated with disease-free survival across 19 analysed solid cancer types (multivariate Cox proportional hazard (Cox-ph) model with significance level $\alpha = 0.05$; Supplementary Data 4, 5). Importantly, this included CAAs that predicted poor or indeed good survival outcome (Fig. 3a, b, Supplementary Data 4 and 5). Interestingly, CAAs that predicted good survival outcome on average did so for more patients than CAAs that predicted poor survival ($p = 0.0002$, Mann–Whitney *U* test; Supplementary Fig. 8). However, the number of identified significant associations for poor survival is 2.4 times higher than for good survival (51 versus 21). Hence, on aggregate, CAAs predicted good and poor prognosis for similar fractions of patients (25% and 23%, respectively).

**88 CAA pairs synergistically predict survival outcome.** We hypothesised that specific CAAs co-occur in tumours more frequently than expected and thereby synergistically predict poor patient outcome. To test this, we first used a probabilistic model originally developed to identify statistically significant pair-wise patterns of species co-occurrence[27]. This method determines the probability that two events co-occur, while accounting for the frequencies of the individual events. We generated matrices of co-occurring CAAs for all 29 solid cancer types to identify CAAs that co-occur at significantly higher frequencies (positive co-occurrence) or lower frequencies (negative co-occurrence) than expected by chance (Fig. 4a, Supplementary Fig. 9, Supplementary Data 6). To better visualise the frequencies, complexities and significance levels, we also generated networks of significant co-occurrences (Fig. 4b, Supplementary Fig. 9) and volcano plots (Fig. 4c, Supplementary Fig. 10). Altogether, this identified 293 negative and 8,373 positive significant CAA co-occurrences across 29 solid cancer types ($q < 0.05$; Supplementary Data 6).

We then assessed whether CAA pairs that co-occurred both frequently and at high statistical significance predict patient survival outcome. This yielded mixed results (Supplementary Fig. 11a–i). For instance, −1p and −19q co-occurred in 30.7% of low-grade gliomas (LGG)—while only 14.5% was expected by chance ($q < 10^{-5}$; Fig. 4a, b)—and this strongly predicted good overall patient survival (log-rank $p = 6.7 \times 10^{-5}$; Supplementary Fig. 11a), a phenomenon that is in fact well-established[13]. On the other hand, in all other datasets the most significant and abundant co-occurring CAAs did not significantly predict patient outcome (Fig. 4c, Supplementary Fig. 11b–i).

Thus, as an alternative, we used the aforementioned unbiased multivariate approach. This highly robust analysis identified 88 co-occurring CAA pairs that were significantly associated with patient survival across 19 solid cancer types (Supplementary

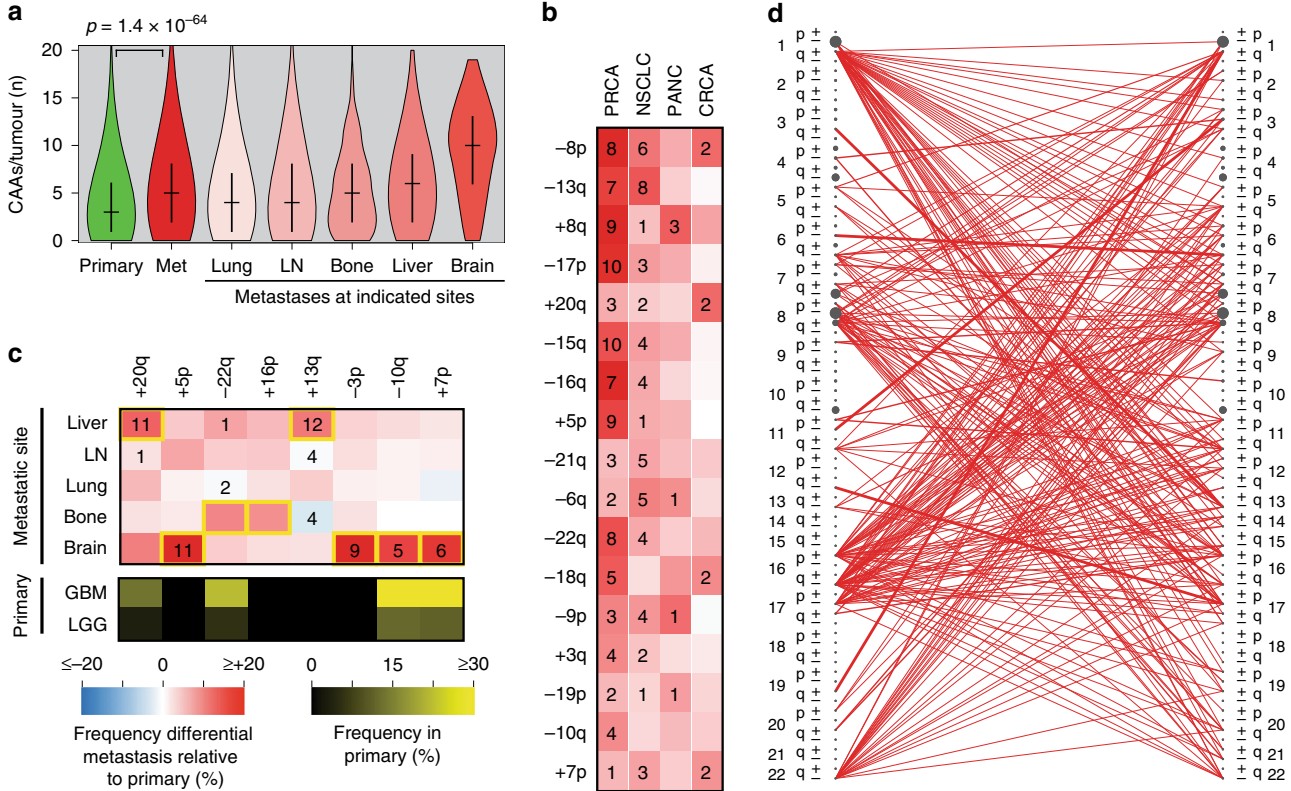

**Fig. 2 CAAs in primary and metastatic samples shape tumour evolution. a** Violin plots comparing CAA burden in primary and metastatic solid tumours. Medians and interquartile ranges are shown. LN, lymph node; Met, metastases at any metastatic site. P value: Mann–Whitney U test. **b** Heatmap of the frequency differential in metastatic disease relative to primary cancers. Numbers in tiles refer to q values, i.e., FDR-adjusted p values of Fisher's exact tests. Empty tiles, not significant ($q > 0.05$); 1, $q < 0.05$; 2, $q < 10^{-2}$; 3, $q < 10^{-3}$, etc. **c** Heatmap as in **b**, but per metastatic site. Yellow boxes highlight a degree of specificity. The black/yellow heatmap below shows the frequencies of indicated CAAs in primary brain cancers. **d** Stochastic tumour evolution model. The network models the order in which CAAs are acquired during breast cancer development. Nodes represent CAAs and their sizes are proportional to CAA frequencies. Edges represent estimated transition probabilities and their thicknesses are proportional to the probabilities. Note that the edges are directed, from left to right. Source data are provided as a Source Data file.

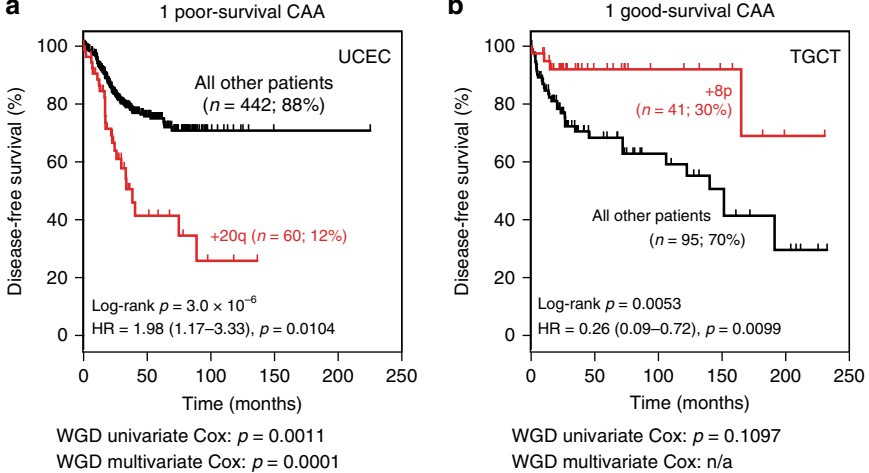

**Fig. 3 Identification of 72 CAAs that predict good or poor cancer patient survival outcome. a**, **b** Examples of Kaplan–Meier curves showing disease-free survival of patients with/without indicated CAA. P values: log-rank test and multivariate Cox proportional hazard analysis. Hazard ratios (HR) and 95% confidence intervals are shown. The effect of whole-genome doubling (WGD) on survival outcome was also assessed in univariate and multivariate analyses. See also Supplementary Data 4 and 5. Source data are provided as a Source Data file.

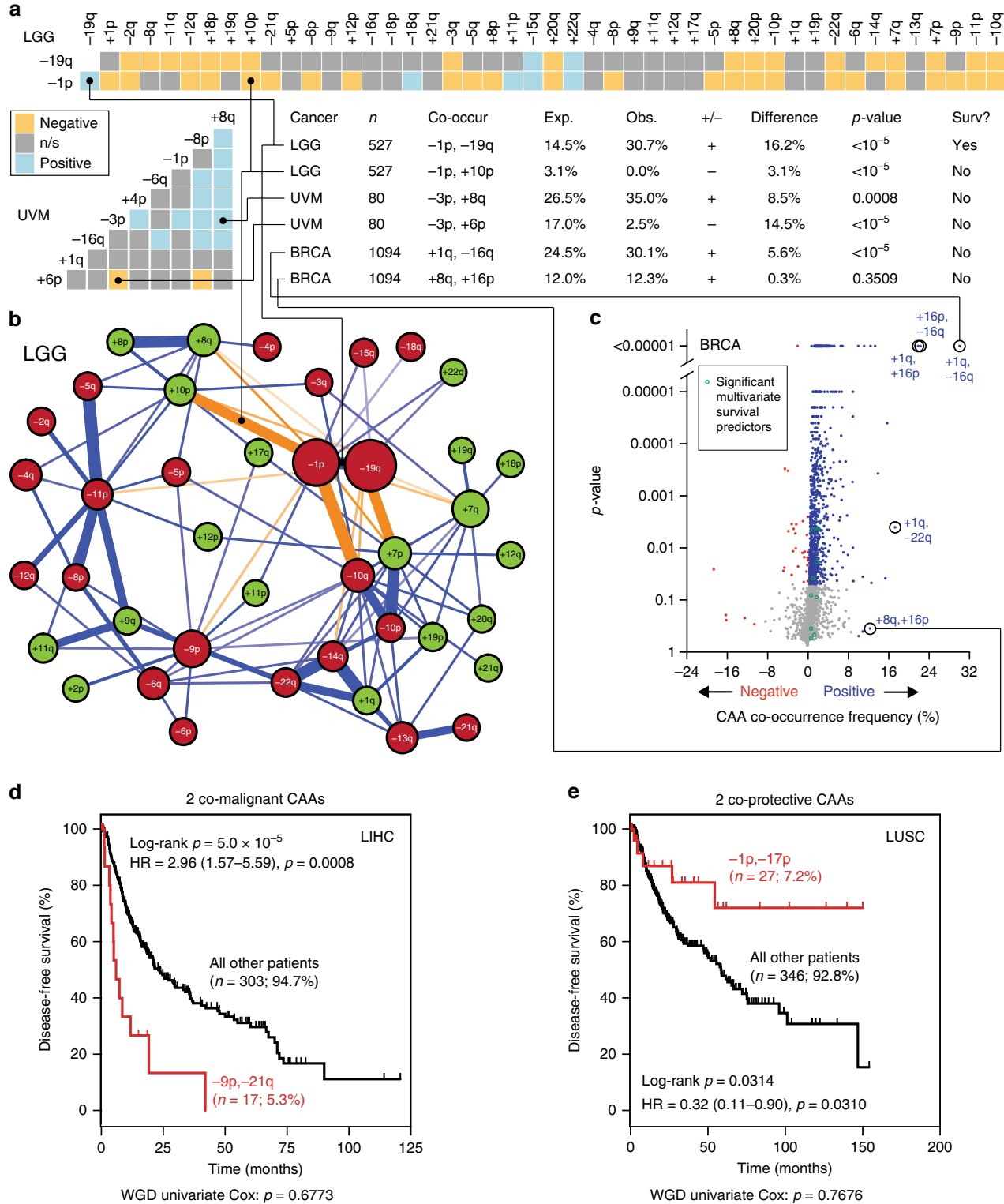

**Fig. 4 Identification of 88 co-occurring CAA pairs that synergistically predict good or poor cancer patient survival outcome. a** Matrices and table of selected results from pan-cancer CAA probabilistic cooccurrence analyses. Tile colours indicate whether CAA combinations occur significantly more ('positive') or less ('negative') frequently than expected. **b** Network of CAA cooccurrences in the LGG dataset. Nodes represent CAAs, including losses (−, dark red) and gains (+, light green). Sizes are proportional to frequencies. Edge colours indicate statistically significant positive (dark blue) or negative (orange) probabilities as in (**a**) with thickness inversely proportional to probability. **c** Volcano plot showing frequencies and probabilities of co-occurring CAAs in the BRCA dataset. Pairs co-occurring at $p > 0.05$ or involving > 10% of patients are shown in blue (positive) or red (negative). Green circles highlight co-occurrences significantly predicting patient survival outcome in multivariate analyses. **d**, **e** Example Kaplan–Meier survival curves of co-occurring CAA pairs predicting poor (**d**) or good prognosis (**e**). Statistics as in Fig. 3a, b. Source data are provided as a Source Data file.

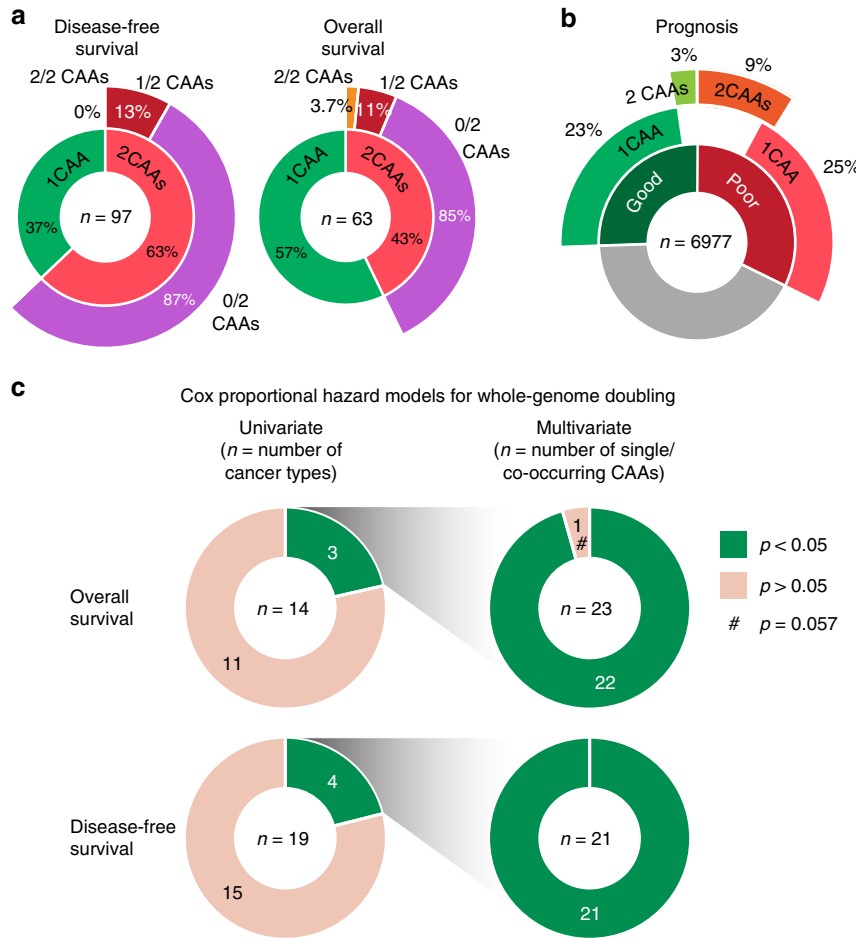

**Fig. 5 Summary of CAA-informed survival outcome, with negligible impact of whole-genome doubling. a** Sunburst charts for disease-free and overall survival. In total, $n = 97$ and $n = 63$ statistically significant individual CAAs ('1 CAA') or CAA combinations ('2 CAAs') were identified. Inner rings indicate fractions of 1 CAA and 2 CAAs. Outer rings indicate how many of the two individually cooccurring CAAs significantly predict patient survival outcome. **b** Sunburst chart showing number of patients analysed ($n = 6,977$) and fractions for whom CAAs predict good (green, left) or poor prognosis (red, right) based on individual ('1 CAA') or co-occurring CAAs ('2 CAAs'). **c** Left: Ring pie charts showing the number of cancer types for which whole-genome doubling (WGD) significantly affects poor overall or disease-free survival in univariate Cox proportional hazard analyses. Right: Ring pie charts depicting the multivariate Cox proportional hazard $p$ values of the 44 single or co-occurring CAAs for which WGD showed $p$ values < 0.05 in univariate analyses (left). These multivariate analyses included WGD as a covariate. Source data are provided as a Source Data file.

Data 4 and 5). These co-occurrences predicted poor or good patient survival outcome and typically occurred at low frequencies, on average involving 9% of patients (Fig. 4c–e, Supplementary Data 7).

Overall, our multivariate survival analyses identified more associations with co-occurring CAAs than with single CAAs (Fig. 5a). However, the latter involved more patients (Fig. 5b). Notably, 86% of significant co-occurring CAAs involved two CAAs that individually were not significant predictors, indicating synergism (Fig. 5a). In addition, a considerable number of CAAs predicted good patient outcome (Fig. 5b) and co-occurring CAAs were predictive for survival in 10% of patients for whom single CAAs were not (Fig. 5b). Together, individual and co-occurring CAAs significantly predicted good or poor survival for 58% of patients (Fig. 5b).

We also assessed the effect of WGD on the survival prognostic power of CAAs. Univariate Cox-ph analyses identified WGD to significantly impact survival outcome in 7 out of our 33 cancer type-survival type combinations (Fig. 5c, Supplementary Data 4 and 5). This involved 44 individual CAAs or co-occurring CAA pairs that we previously identified as significant survival predictors (Supplementary Data 4 and 5). For these, inclusion

of WGD as a co-variate in multivariate analyses yielded a Cox-ph $p$ value > 0.05 in only 1 of these 44 cases (specifically, $p = 0.057$; Fig. 5c, Supplementary Data 4 and 5). Thus, WGD has a negligible effect on the patient survival prognostic power of CAAs.

**CAAs are independent predictors of drug response.** We investigated relationships between CAAs and chemotherapeutic drug response. First, CAA burden is typically positively associated with increased predicted pathologic complete response (pCR) to preoperative paclitaxel and fluorouracil-doxorubicin-cyclophosphamide (T/FAC) chemotherapy, as determined using a pharmacogenomic predictor[28] (Fig. 6a, Supplementary Data 8 and 9).

This encouraged us to comprehensively investigate if CAAs could predict response to individual chemotherapeutic drugs. To this end, we utilised data from the Sanger Institute's Genomics of Drug Sensitivity in Cancer (GDSC) project, which comprehensively profiled the landscape of responses of 1,001 human pan-cancer cell lines to 265 anti-cancer drugs[21]. To ensure that the predictive potential of CAAs can be directly compared to

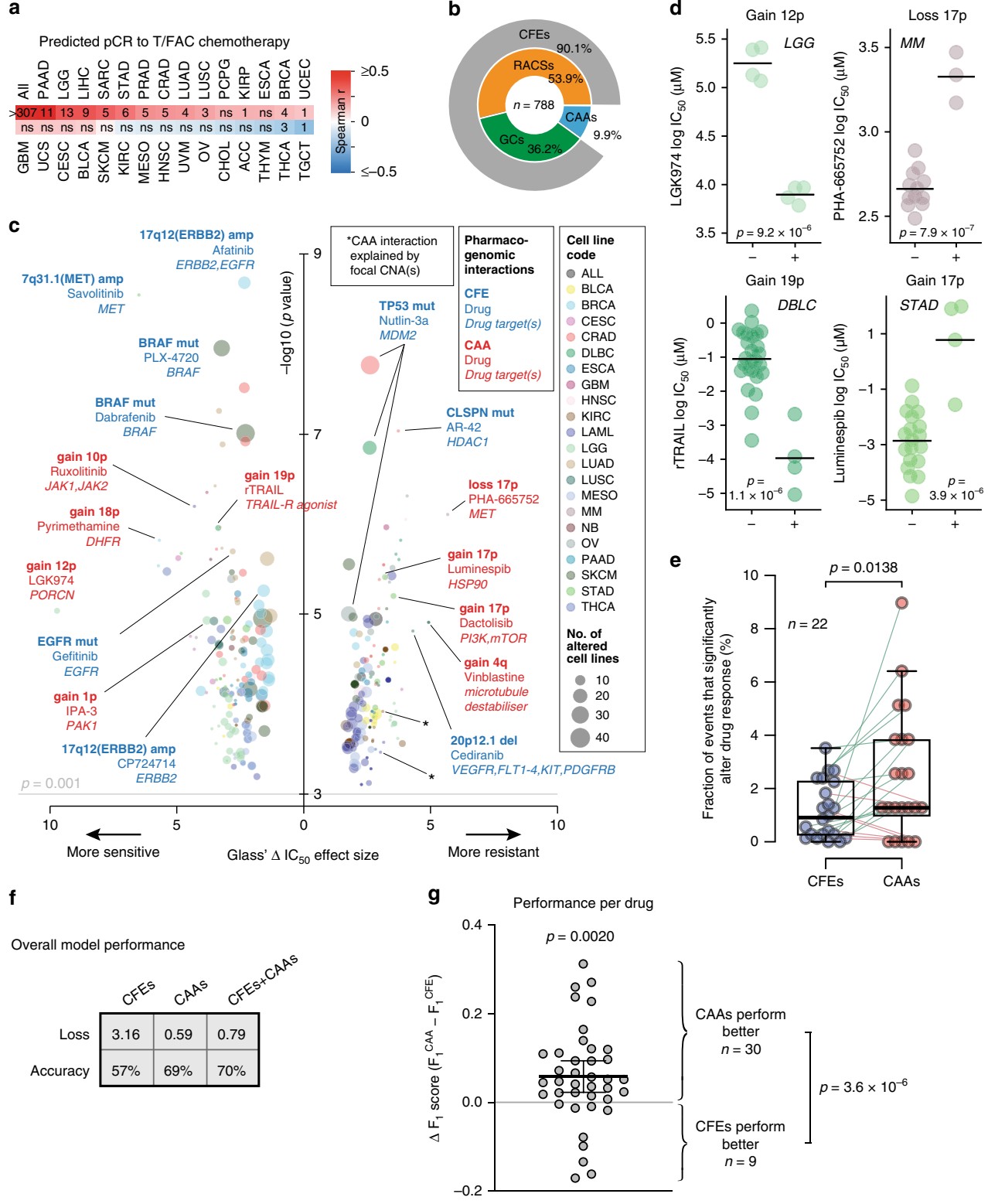

previously established pharmacogenomic predictors[21], we used the same machine learning pipeline, specifically elastic net regression. Our model included the following: (a) 710 established pan-cancer pharmacogenomic features, referred to as cancer functional events (CFEs), including mutations in 285 high-confidence cancer genes (GCs) and 425 recurrently copy number-altered chromosomal segments (RACSs) (Fig. 6b)[21,29]; (b) the 78 CAAs that we determined here (Fig. 6b); (c) 386,293 IC$_{50}$ values of an expanded panel of 453 anti-cancer drugs, including 265

previously reported[21] and 188 additional drugs (see Methods); and (d) 988 cancer cell lines. This identified 365 significant CFE/CAA-drug interactions (at $p < 0.001$ and FDR < 0.25) with a Glass's Δ IC$_{50}$ effect size > 1.0 across 22 cancer types (Fig. 6c, d, Supplementary Data 10). Of these 365 interactions, 301 involved CFEs and 64 involved CAAs (Supplementary Data 10). CAAs consistently ranked among the top 50 features most frequently associated with drug response, alongside known CFE-drug interactions, both at the individual tissue and pan-cancer levels

**Fig. 6 CAAs shape drug response and outperform other genomic events in response prediction. a** Heatmap of Spearman correlations between CAA burden and a pharmacogenomic predictor of pathologic complete response (pCR) to preoperative paclitaxel and fluorouracil-doxorubicin-cyclophosphamide (T/FAC) chemotherapy. Numbers in tiles show $q$ values, i.e., FDR-corrected significance from Fisher's exact tests, as Fig. 2b. Ns, not significant ($q > 0.05$). **b** Sunburst plot showing the distribution of pharmacogenomic alterations used in our machine learning model, including 285 high-confidence cancer genes (GCs), 425 recurrently copy number-altered chromosomal segments (RACSs), collectively referred to as cancer functional events (CFEs), and 78 CAAs. **c** Bubble volcano plot showing how specific CFEs and CAAs alter response to anti-cancer drugs, as determined by elastic net regression. The impact is shown as Glass' $\Delta \log_{10}(IC_{50})$ effect size between cell lines with and without the alteration. Bubble colours correspond to cancer types. Bubble sizes are proportional to the numbers of cell lines. Selected CFEs and CAAs are highlighted in blue and red, respectively. Two CAA-drug interactions can be explained by focal CNAs (*). **d** Beeswarm plots showing the extent to which several CAAs significantly increase drug resistance or sensitivity. Cell lines negative ($-$) and positive ($+$) for indicated CAAs are shown. Horizontal lines represent the mean $IC_{50}$ value. See Supplementary Data 10 for full data. **e** Tukey boxplot showing the fractions of CFEs or CAAs that significantly alter drug response. Shown are the medians with interquartile ranges and all data points. Lines connect the respective fractions for each of the 22 cancer types. $P$ value: Paired Wilcoxon signed-rank test. **f** Summary table showing the performance of machine learning models based on CFEs alone, CAAs alone and CFEs and CAAs combined. **g** Beeswarm plot showing the differences between the $F_1$ performance scores from the CFE-based and the CAA-based models for each of the tested 39 drugs. Mean and 95% confidence interval are shown. $P$ values: one sample $t$-test (top), Fisher's exact test (right). Source data are provided as a Source Data file.

(Supplementary Data 10). On aggregate, we identified 31 distinct CAAs that robustly alter response to 56 chemotherapeutic drugs across cell lines representing 17 cancer types.

It is possible that some CAA-drug interactions do not represent new information if focal CNAs on the respective chromosome arms already predict such interaction. This was the case for 2 of the 64 CAA-drug interactions (4q loss and 4q13-q31 focal deletions: AURK inhibitor resistance in KIRC (which may be linked to loss of the gene encoding the Aurora A interactor HNRNPD[30]); 17p loss and 17p11-p13 focal deletions: MCL-1 inhibitor resistance in LAML (which may be linked to TP53 loss, as p53 interacts with MCL-1[31])) (Supplementary Data 10). Using Fisher's exact tests, additional systematic in-depth analysis of all possible $_{788}C_2 = 310,078$ co-occurring CFEs/CAAs, including co-occurring focal CNAs, on the 988 cell lines (Supplementary Data 11) did not identify any additional such cases, even at a relaxed threshold of raw $p$ value $< 0.05$ without FDR correction (Supplementary Data 12). Thus, our high-dimensional analyses indicate that individual CAAs can be used as independent predictors of chemotherapeutic drug response in cell lines.

**Potential synthetic lethal interactions of drug sensitivity.** We systematically tested whether, for each of the 28 cancer types, as well as pan-cancer, any of the possible $\binom{788}{2} = 310,078$ co-occurring CFE/CAA pairs are significantly more frequently associated with drug sensitivity than with drug resistance, or vice versa, using Fisher's exact tests. Using cut-offs of $p < 0.05$ and FDR% $< 0.001$, we identified 1,024 potential synthetic lethal interactions, i.e., co-occurring events associated with increased sensitivity, and 89 potential synergistic resistance interactions (Supplementary Data 11).

**CAAs outperform other genomic events in predicting drug response.** We found that, corrected for the number of input alterations, CAAs are involved in drug interactions nearly twice as often as CFEs across 22 cancer types ($p = 0.0138$, paired-sample Wilcoxon test; $p = 7.9 \times 10^{-7}$, Chi-square test) (Fig. 6e, Supplementary Fig. 12a, Supplementary Data 10).

To assess how well CAAs could predict drug response compared to CFEs, we developed a deep neural network model to predict drug sensitivity and used binary resistant/sensitive calls for the above 988 cell lines and 265 anti-cancer drugs, as previously reported[21]. The vast majority of these calls were 'resistant'. Hence, to avoid mostly identifying genomic events associated with drug resistance, we used drugs for which at least 15% of the cell lines had sensitive calls (see Methods).

Accordingly, our deep neural network models included 39 drugs in 971 cell lines. We used these data to train and 5-fold cross-validate three independent pan-cancer deep neural network models, which were respectively based on CFEs only (710 CFEs), CAAs only (78 CAAs) and the combination of CFEs and CAAs (788 features) (Fig. 6b).

The validation accuracy of the CFE-based model was 57%. With an accuracy of 69%, the CAA-based model performed considerably better (Fig. 6f, Supplementary Fig. 12b, c). While, with an accuracy of 70%, the model based on both CFEs and CAAs performed best, this was a negligible improvement compared to the model based on CAAs alone (Fig. 6f, Supplementary Fig. 12d). For more in-depth evaluation of the models, we carried out performance analyses for each drug. This revealed that CAAs performed better than CFEs for 30 of the 39 drugs, as assessed by higher $F_1$ scores (Fig. 6g, Supplementary Data 13). This represents a statistically significant difference ($p = 3.6 \times 10^{-6}$, Fisher's exact test, Fig. 6g). Notably, for drugs where CFEs performed better, there is often a well-established CFE-drug interaction, such as for lapatinib and focal gain of ERBB2/EGFR (Supplementary Data 10, 13). We also found that on average, the difference in $F_1$ scores showed a significant bias towards CAAs ($p = 0.0020$, one-sample $t$-test; Fig. 6g). Thus, 78 CAAs both quantitatively and qualitatively outperform 710 well-established CFEs and improve the power of predicting drug response based on pharmacogenomic parameters.

**Discussion**
Herein, we performed comprehensive pan-cancer analyses of chromosome arm-level aneuploidies (CAAs), a common consequence of genomic instability in cancer cells[1–5]. Several studies have previously studied CAAs across multiple cancer types[15–17]. These revealed that CAAs are common in cancers, yet they are tumour type-specific, and CAA burden is associated with TP53 mutations, cell proliferation, cell cycle gene expression and low levels of tumour-infiltrating immune cells. In contrast, our study utilises CAAs to identify aspects of tumour evolution, metastasis, patient survival and chemotherapeutic drug response.

Our analyses provide thus far unknown broad and specific insights into tumour evolution. Solid cancers show a considerably higher CAA burden than haematological cancers. This may reflect the fact that haematological cancer development is largely driven by translocations, or that TP53 mutations are less prevalent in these cancers[16,17,32,33]. Another discriminating feature between haematological and solid cancers is their opposing bias towards gain and loss of chromosome arms, respectively. Importantly, however, in solid cancers the bias towards arm loss is a function of the total CAA burden. Indeed, these cancers show

a preference towards chromosome arm gain when the CAA burden is low, akin to haematological cancers, but preferential loss when the CAA burden is high.

We propose a tumour evolution model in which solid cancers initially preferentially gain chromosome arms, whereas they preferentially lose chromosome arms later during development (Fig. 1f). Aside from the above observation, two lines of evidence further support this model. First, the number of CAAs significantly increases with more advanced stages of disease, in particular from stages I to II and II to III (Fig. 1g, Supplementary Fig. 4e). Second, the fraction of CAA-positive solid tumours with more chromosome arm gains than losses is significantly higher in stage I tumours than in tumours at advanced stages. Interestingly, even stage I tumours show a slight bias towards chromosome arm loss. This suggests that the bias towards gain may be restricted to the earliest stages of solid tumour development. Consistently, we observe this bias only in tumours with 1 and 2 CAAs and the median CAA burden of stage I tumours is 4. This suggests that early during tumorigenesis, the gain of oncogenes may be more important than the loss of tumour suppressor genes, although this requires overcoming oncogene-induced senescence[34,35]. Alternatively, or additionally, instantaneous hemizygous deletion of multiple genes, including essential genes, may reduce the fitness of incipient cancer cells[17,36–38]. Such losses may be better tolerated in a 4n background, as suggested by our observation that WGD predisposes to increased CAA burden.

Our study demonstrates the value of CAAs for cancer diagnostic and, potentially, therapeutic purposes in various ways. We identified 72 specific CAAs and 88 co-occurring CAA pairs that significantly predict good or poor patient survival outcome in multivariate analyses and these respectively involve 48% and 12% of patients. We noted that there is very low patient overlap between these percentages. Hence, for 58% of all patients, good or poor survival can be predicted using CAAs alone. Additionally, increased CAA burden predicts increased rate of metastasis, in particular to brain, and several CAAs show specificity for metastatic sites, similar to recent observations[39]. WGD has a negligible effect on the prognostic power of CAAs.

Loss of chromosome arm 3p was previously identified as common in renal cell carcinoma and in squamous cancers of the lung, head and neck, oesophagus and cervix[17,19,20]. However, 3p loss does not promote proliferation of primary human lung cells[17]. This suggests that additional aberrations are required. This could include *TP53* mutations in some—but not all—cancers, as these strongly correlate with SCNA burden and aneuploidy, and p53 protects against structural aneuploidies that originate in mitosis[17–19,40]. Alternatively, 3p loss may concomitantly require other specific SCNAs. We identify 5q loss as a strong candidate, because it significantly cooccurs with 3p loss in 11 cancer types, including all five of the aforementioned cancer types with frequent 3p loss ($q < 10^{-5}$; Supplementary Data 6). This includes frequencies of up to 2.4-fold higher than expected in lung cancers, involving 38% of lung squamous cell carcinoma patients (Supplementary Data 6). Also, our multivariate survival analyses identified 3p loss as a significant prognostic predictor for poor survival in 1 of 19 analysed cancer types (UCEC, Supplementary Data 4 and 5). Yet, cooccurrence of 3p loss with other CAAs, including 5q loss, significantly predicts poor patient survival for three cancer types (Supplementary Data 5). This example demonstrates the value of our integrative approach utilising CAA frequencies, cooccurrence analysis and multivariate survival analysis for clinical prognostic purposes.

Finally, we performed machine learning drug response modelling to identify CAA-drug interactions. We included a broad range of 788 genomic features. Among them were mutations in high-confidence cancer genes and recurrently copy number-

altered chromosomal segments, collectively CFEs, as well as CAAs. Our model identified previously identified CFEs that significantly predict drug response[21,41]. In addition, we identified 31 specific CAAs, 30 of which independently increase resistance or sensitivity to 54 chemotherapeutic drugs across cell lines representing 17 cancer types. Only 2 of the total 64 CAA-drug interactions could be explained by focal CNAs.

CAAs considerably outperform CFEs in predicting drug response. This is remarkable, because over 90% of the 788 features were CFEs and less than 10% were CAAs. In addition, the 710 included CFEs were preselected, only comprising high-confident and recurrent events, whereas all 78 CAAs were included without preselection. However, with a combined accuracy of 70%, there is room for improvement of the predictive performance. Our model was exclusively based on pharmacogenomic features. Thus, we anticipate that inclusion of transcriptomic or proteomic parameters could further improve performance. In addition, a potential problem in machine learning involves the presence of confounding factors, known or unknown, which can affect model performance[42]. Such factors could include cancer subtypes, which our models did not account for. Thus, inclusion of subtype and application of confounder control methods could improve model performance[43].

The functional consequences of CAAs are not always easily understood. Two of the identified 64 CAA-drug interactions could be explained by focal CNAs, suggesting involvement of *HNRNPD* and *TP53* loss (see above). Other CAA-drug interactions could not be explained by co-occurring focal CNAs on the same chromosome arm, even if distant from each other. Thus, more complex interactions exist, potentially synthetic lethal events involving three or more loci. While in vitro studies are required to understand the full consequences of individual CAAs[17], our work does provide some clues. For example, 17p loss increases resistance to seven different drugs in leukaemia (LAML) and five of these target cell cycle/mitotic regulators (KIF11, CDK2/7/9, WEE1, PLK1, microtubules) (Supplementary Data 10). This links 17p loss to resistance to cell cycle inhibitors. This may well involve a complex interaction with *TP53*, as it is located at 17p13.1, while *TP53* loss (or mutation) alone is not predictive (Supplementary Data 10).

We also highlight that context matters at several levels. This applies to the broad genomic context, as evidenced by our analyses involving individual CAAs, which are akin to a large number of focal CNAs, and co-occurrence analyses, including in the context of patient survival and potential synthetic lethal interactions. There are also vast cancer type-specific differences. Additionally, the tumour microenvironment is complex, involving clonal and sub-clonal aberrations, as well as other cell types, including tumour-infiltrating lymphocytes, whose abundance inversely correlates with aneuploidy[16,17]. In this light, our observation that CAAs strongly predict drug response may lay a foundation for pre-clinical studies, involving validation in mouse xenograft, patient-derived cancer organoid (PDO) or xenograft (PDX) models[44,45]. This will be critical, because even though cell lines typically well represent genetic and genomic somatic alterations found in tumours[21], PDX models in particular much better mimic the complexities that exist in the tumour microenvironment[45]. Taken together, our findings can be a starting point for pre-clinical studies and hence have the potential to ultimately advance precision oncology.

## Methods

**Ethical compliance**. All data from human subjects were obtained from public resources and access was either unrestricted or restricted (see also Data Availability Statement below). All subject data were non-identifiable and all participants provided written informed consent and complied with ethical regulations as

determined by the Ethical Boards reported: TCGA: https://www.cancer.gov/about-nci/organization/ccg/research/structural-genomics/tcga/history/policies; MSK-IMPACT[26]; METABRIC[22]. In addition, we received Institutional Human Research Ethics Approval from the Medical Research Ethics Committee (MREC) at the University of Queensland.

**Chromosome arm-level aneuploidy profiling.** Pre-processed SNP6.0 array segmental copy number data were available for TCGA and MSK-IMPACT. For METABRIC and GDSC, raw CEL files were downloaded. These data were processed in Python (version 3, www.python.org) using the TCGA pipeline to provide coherence between datasets. This pipeline uses raw SNP6 CEL files as input and generates segmented copy number calls (log2 ratios) for each patient sample or cell line. Specifically, raw data were processed as follows. Signal intensities were calibrated using *SNPFileCreator_SNP6*. Genotype calls were computed using *Birdseed[46]*. Signal intensities were converted into copy number calls using *CopyNumberInference*. Copy number noise was calculated using *CopyNumberNoise* and reduced first by removing outlier probes using *RemoveCopyNumberOutliers* and second by subtracting variation observed in normal samples using *TangentNormalization*. Finally, contiguous chromosome regions with log2 ratio segment means were obtained using the package *DNACopy* in the *R* statistical environment (R Core Team, Vienna, Austria). SNP6 copy number segments were considered lost, gained or unchanged with respect to the ploidy status of the sample. Thus, copy numbers were called independent of whole-genome doubling (WGD) status, unless indicated otherwise (see also section Whole-genome doubling analyses below). TCGA, MSK-IMPACT, METABRIC and GDSC data were post-processed with GISTIC2.0 using a threshold of >0.2 for amplification and <−0.2 for deletion[47]. Segmental somatic copy number calls were extracted along with frequencies by cancer type. Somatic copy number alterations (SCNAs) were determined by subtracting germline copy number calls from tumour copy number calls aligned to Human Genome Build GRCh37/hg19 using Python (version 3). For each segment in the SCNA file, if the segment intersected the centromere positions (as downloaded from UCSC genome browser for GRCh37/hg19 human genome reference build; Supplementary Data 14), it was discarded. For each sample in the SCNA file, total segment lengths (regardless of direction or segment mean) were summed for each chromosome arm. For each chromosome arm the length of amplification or deletion (with |segment men| > 0.2) was summed. Fractions reported were lengths of amplification/deletion divided by total length of segments (per chromosome arm). Chromosome arm-level SCNAs/aneuploidies (CAAs) were called by scoring individual chromosome arms as gained or lost if ≥0.9 of the arm was gained or lost. These cut-offs allow for background noise, heterogeneity and deviating focal copy number aberrations[14,15,24]. This did not distinguish between heterologous loss, homozygous loss or both, which were analysed as one group. This method was then subjected to technical and biological validation, as described below.

**Technical and biological validation of CAA frequencies.** The above method was technically validated on the breast cancer datasets by comparing our CAA frequencies ($n = 1094$) to those recently reported by Taylor et al.[17] ($n = 1048$), who also determined CAA frequencies in TCGA samples using a similar method, and to CAA frequencies determined from TCGA whole-genome (WG) microarray data ($n = 1081$). For the latter, thresholded gene-level copy number estimates were used, as determined by GISTIC2 and the TCGA firehose pipeline (https://gdac.broadinstitute.org)[47], and these were mapped to the human genome using UCSC xena HUGO probeMap. Consistent with our method using SNP6 array data, CAAs were called if ≥90% on each arm were gained or lost. For biological validation, our CAA frequencies of the largest TCGA dataset (BRCA, $n = 1094$) were compared to the CAA frequencies of the METABRIC breast cancer dataset ($n = 1980$)[22]. Finally, whole-genome sequencing data from the ICGC/PCAWG project were used for validation (n = 214). For this, PCAWG-11 consensus segmented copy numbers were used as determined by the PCAWG-11 working group: "These profiles contain clonal copy number for nearly the complete genomes and are the result of a bespoke procedure that combines output from 6 different copy number callers: ABSOLUTE, ACEseq, Battenberg, CloneHD, JaBbA and Sclust. PCAWG-11 working group first ran all methods across all samples with the consensus SVs included and applied an algorithm across the segmentations to obtain consensus breakpoints. With these mandatory breakpoints the methods were rerun without calling any additional breakpoints. Samples for which the methods disagreed on the ploidy have gone through an adjustment algorithm that applies various ploidy adjustments in order to maximise the agreement and were assessed through a rigorous review procedure. After obtaining consensus on the ploidy, segments are considered individually to assign copy number states: Clonal agreement (3 stars), majority vote agreement and agreement after rounding subclonal copy number (2 stars) and a call from the best method on that sample (1 star). This finally yields a complete, clonal copy number profile." Subsequently, the segmented copy numbers were subjected to the same pipeline described above for SNP6 data to call the CAAs.

**Statistical analysis.** False discovery rate (FDR), yielding q values, was applied to account for multiple-hypothesis testing[48]. Where p or q values are summarised, abbreviations are as follows: ****<0.0001; ***<0.001; **<0.05; n/s, not statistically

significant. Alternatively, where indicated, q values are summarised as follows: ns, not significant ($q > 0.05$); 1, $q < 0.05$; 2, $q < 10^{-2}$; 3, $q < 10^{-3}$, etc. All statistical tests were two-tailed, unless specifically indicated otherwise. Other statistical methods are explained in detail below.

**Whole-genome doubling analyses.** As described above, CAAs were determined with respect to the ploidy status of the sample and hence independent of whole-genome doubling (WGD) status. Whether tumours had undergone WGD was assessed using the ABSOLUTE algorithm[49] (TCGA samples) or called if more than half of the autosomal genome had two or more copies of the more frequent (maternal or paternal) allele[25] (MSK-IMPACT samples). To determine the relationship between CAAs and WGD, various analyses on all samples, which included both WGD- and WGD+ samples, were either repeated for WGD- and WGD+ samples separately, as indicated, or WGD status was included as a covariate in multivariate analyses (see also Patient survival analyses section below). Where WGD status is not specified, analyses included both WGD− and WGD+ samples.

**CAA burden and response to selected chemotherapeutic drugs.** The pathologic complete response (pCR) to preoperative paclitaxel and fluorouracil-doxorubicin-cyclophosphamide (T/FAC) chemotherapy was assessed using a pharmacogenomic predictor that was previously described[28] (Supplementary Data 9). Per cancer type, the extent to which the number of CAAs per tumour correlated with this parameter was determined by Spearman's rank correlations. This rendered regression coefficients and p values. False discovery rate (FDR)-adjusted q values were calculated to correct for multiple-hypothesis testing[48] (Supplementary Data 8). Only q values smaller than 0.05 were considered statistically significant.

**Probabilistic computation of intra-tumour gain:loss ratios.** For each of the intra-tumour categories 'gain > loss', 'loss > gain' and 'gain = loss', expected CAA frequencies in haematological and solid cancers are a function of the total number of CAAs per tumour sample. Let $a$ be the total number of CAAs in a sample and $f_a$ the observed frequency of samples with $a$ CAAs. Equation (1) was used to calculate the expected frequency of the number of samples with more chromosome arm gains than losses for a given $a$.

$$\mathbb{E}_{\text{gain}>\text{loss}}(f_a) = \begin{cases} \left(\dfrac{1}{2} - \dfrac{\binom{a}{a/2}}{2^{a+1}}\right) \times f_a & \text{if } a = \{2k : k \in \mathbb{Z}_+\} \\ \dfrac{1}{2} \times f_a & \text{if } a = \{2k+1 : k \in \mathbb{Z}_{\geq 0}\} \end{cases} \quad (1)$$

In addition, $\mathbb{E}_{gain=loss}(f_a) = \mathbb{E}_{gain>loss}(f_a)$, while $\mathbb{E}_{gain=loss}(f_a)$ was calculated using Eq. (2).

$$\mathbb{E}_{\text{gain}=\text{loss}}(f_a) = \begin{cases} \dfrac{\binom{a}{a/2}}{2^a} \times f_a & \text{if } a = \{2k : k \in \mathbb{Z}_+\} \\ 0 & \text{if } a = \{2k+1 : k \in \mathbb{Z}_{\geq 0}\} \end{cases} \quad (2)$$

Overall expected frequencies are a function of both $a$ and $f_a$, as specified in Eq. (3).

$$\mathbb{E}_{\text{overall } i}(f_i) = \sum_{a=1}^{a_{\max}} \mathbb{E}_i(f_i) \text{ with } i \in \{\text{gain} > \text{loss}, \text{loss} > \text{gain}, \text{gain} = \text{loss}\} \quad (3)$$

**Primary and metastatic samples.** SNP6 array data from the MSK-IMPACT study were obtained and processed as described above to identify CAAs in each of the 10,202 samples (Supplementary Data 2). These were used to compare CAA frequencies and burden between primary and metastatic samples[26]. Fisher's exact tests and FDR-adjusted q values were used to determine whether frequencies of individual CAAs differed between tumour types or primary and metastatic sites. Mann–Whitney $U$ tests and FDR-adjusted q values were used to assess if CAA burden differed between tumour types or primary and metastatic sites.

**Stochastic tumour evolution modelling.** For the TCGA-BRCA dataset, a multi-level tree was created in which each node $v \in V$ represents a CAA 'karyotype': a one-dimensional array $A$ with 78 elements, one for each possible CAA, which can be absent, '0', or present '1' (Supplementary Fig. 7a). Let $G = (V, E)$ be such directed tree of a given dataset, in which $E$ represents the edges. Level 0 is represented by a single node of a 'normal karyotype' without CAAs, $A_0 = [0, \ldots, 0]$. Each subsequent level $k$ is represented by all nodes that each correspond to every unique 'karyotype' with exactly $k$ CAAs. Each edge $e \in E$ is directed from a node on level $k$ to a node on level $k + 1$, such that the 'karyotype' of the node on level $k + 1$ equals that of the origin node on level $k$ plus one additional CAA (Supplementary Fig. 7b). Each node has an associated frequency, referring to the number of times the corresponding 'CAA karyotype' occurred in the dataset. This tree was used to estimate transition probabilities and model the sequential acquisition of CAAs during tumour evolution. The probability of acquiring CAA $x$

before CAA $y$, i.e., the transition $\{x \rightarrow y\}$, was estimated using Eq. (4).

$$P_{(x \rightarrow y)} = \frac{W(x \rightarrow y)}{\sum_{y \in A} W(x \rightarrow y) + \sum_{y \in A} W(x \rightarrow \neg y)} \qquad (4)$$

Here, the numerator $W(x \rightarrow y)$ is the total *unnormalized* weight of the transition, i.e., the sum of karyotype frequencies at all end nodes of paths that at the starting node include CAA $x$ but not CAA $y$, and at the end node have attained $y$. The denominator is the sum of *unnormalized* weights of all possible transitions starting at a node with CAA $x$ and ending at a node with or without $y$ (Supplementary Fig. 7b). This method was used to estimate the probabilities of all transitions. This generated a $78 \times 78$ transition probability matrix for the dataset (Supplementary Data 3). Finally, these probabilities, along with all CAA frequencies in the dataset, were used to generate a network graph (Fig. 2d).

**Patient survival analyses.** Both disease-free survival and overall survival data were extracted from the TCGA clinical files. For univariate survival analyses, log-rank Mantel-Cox tests were conducted using GraphPad Prism software, version 7 (GraphPad Software, La Jolla, CA, USA) to determine whether differences were statistically significant. The significance level was set to $\alpha = 0.05$. Multivariate survival analyses were performed per cancer type using Cox proportional hazard regression with a path-wise algorithm on single CAAs (all included) and co-occurring CAAs (included if identified in ≥5 patients) with clinical stage and age at diagnosis as covariates in the model[50,51]. For this, the *R* packages *glmnet* and *coxnet* were used[50,52] (within R version 3.6). These algorithms use a cyclical coordinate descent, computed along a regularisation path[50], which attempts to fit a Cox model that has been regularised by a lasso penalty (ℓ1), after which a cross validation is performed[53]. The maximum number of iterations was set to 1000, as the data was relatively highly dimensional, therefore more iterations were required for convergence. The optimal regularisation parameter $\lambda$ and cross-validated error plot were obtained once the model had been fitted. The covariates that the model chose to be contributing were then taken to fit a new Cox proportional hazards model and the obtained Cox $p$ values were adjusted for multiple-hypothesis testing (type I error) using Bonferroni correction. WGD status was included as a covariate only if it was predicted to significantly ($p < 0.05$) contribute to overall or disease-free survival for the respective cancer types, as determined by univariate Cox proportional hazard regression. Univariate and, if applicable, multivariate significance levels of WGD for all cancer and survival types are provided in Supplementary Data 4 and 5.

**Pairwise probabilistic cooccurrence modelling.** Probabilistic modelling of pairs of CAAs co-occurring in the same tumour sample was based on a previously described model[27,33]. This model is summarised in Eq. (5).

$$p_k = \frac{\binom{N}{k} \times \binom{N-k}{N_2-k} \times \binom{N-N_2}{N_1-k}}{\binom{N}{N_2} \times \binom{N}{N_1}} \qquad (5)$$

Herein, $p_k$ is the probability that CAA 1 and CAA 2 co-occur in $k$ samples out of a total number of $N$ samples, given that CAA 1 occurs in $N_1$ samples and CAA 2 occurs in $N_2$ samples. Thus, the model accounts for the frequencies of each of the two individual CAAs, as well as the total cohort sample size. Cooccurrence matrices, networks and volcano plots were generated in the *R* programming environment (R Core Team, Vienna, Austria).

**Identification of pharmacogenomic interactions.** To identify CFE- and CAA-drug interactions, the multiple input genomic features (CFEs, CAAs) were correlated with the drug response outcome features ($IC_{50}$ values) at the levels of individual cancer types, as well as at the pan-cancer level, using elastic net regularisation as an approach with automatic feature selection using penalised regression. For this, the Python package *GDSCTools*[54] was used, which is available via http://github.com/CancerRxGene/gdsctools. The model was trained with ℓ1 and ℓ2-norm regularisation of the coefficients and the function was minimised using Eq. (6).

$$\frac{1}{2N} \|Y_d - Xw\|_2^2 + \alpha\rho\|w\|_1 + \frac{\alpha(1-\rho)}{2}\|w\|_2^2 \qquad (6)$$

Here, $Y_d$ represents the $IC_{50}$ values for a given drug in all cell lines and $X$ comprises the genomic features (CFEs and CAAs) of these cell lines[55]. The combination of ℓ1 and ℓ2 penalties was controlled by the mixing parameter $\rho$ which was fixed to $\rho = 0.5$ for this analysis. Equation (6) enables learning a sparse model in which few of the weights are non-zero. The $\alpha$ parameter was optimised in the following way. A range of $\alpha$ parameters was scanned to select the best $\alpha$. For each drug, $\alpha$ was tuned with $\alpha \in [0, 1]$ and equal increments of 0.01, wherein each $\alpha$ yielded a concordance index. The concordance index for which the errors were minimum was used. The model was trained on 80% of the data. The remaining 20% served as test data. To avoid over-fitting, the training data were randomly split into 5 equal parts to enable performing 5 times 5-fold cross-validation on the data within the training set. The performance measure used was the average of the values computed on the 5 models. Finally, the model learned was validated on the test data. The metric used to select the best model was the Pearson correlation between predicted and observed $IC_{50}$ values.

**Interactions between drugs and co-occurring genomic features.** Potential synthetic lethal and synergistic resistance pharmacogenomic interactions, as well as co-occurring focal copy number alterations (CNAs) that together might explain CAA-drug interactions, were identified using a systematic co-occurrence approach using the binary drug resistance/sensitivity calls that were previously reported[21]. For each possible $_{788}C_2 = 310,078$ co-occurring CFEs/CAA combination, including co-occurring focal CNAs, a two-sided Fisher's exact test was performed to assess if cell lines with the co-occurring events were significantly more frequently sensitive or resistant than the cell lines without the co-occurring events. Only combinations with significant $p$ values ($p < 0.05$) were stored (Supplementary Data 11). This analysis was performed 23 times, per cancer type, i.e., on the cell lines representing the 22 cancer types listed in Fig. 6c, as well as 'pan-cancer'. Potential synthetic lethal and synergistic resistance pharmacogenomic interactions were called if $p < 0.05$ and FDR% < 0.001. To determine whether co-occurring focal CNAs could explain CAA-drug interactions, such focal CNAs needed to involve segments on the same chromosome arm as the CAA, co-gain or co-loss in the same direction as the CAA and involve the same drug and the same cancer type as the identified CAA.

**Machine learning models for predicting drug response.** GDSC data were downloaded and processed as described above to determine CAAs in each cell line. Additionally, for each cell line, the mutation status of high-confidence cancer genes (GCs), the copy number status of recurrently copy number-altered chromosomal segments (RACSs) at the pan-cancer level, as reported by the GDSC project (see above), and CAAs were considered.

Importantly, across cell lines, the vast majority of calls were "resistant" and a minority of calls were "sensitive". Artificial balancing of the drug response data assumes equal probabilities of finding resistance and sensitive cell lines. Since the latter was not the case, here, artificial balancing of the data would render unreliable models that would over-predict the minority class. Hence, instead an established under-sampling technique was adopted without introducing synthetic examples in the data[56]. This method indicates that for reliable prediction of drug sensitivity, for every drug at least 10% of the calls need to be sensitive. Furthermore, in order to have this minimum number of sensitive cell lines per drug so that there are sufficient examples to split between training and test sets, this worked out to be at least 15% sensitive cell lines for each drug. There were 39 drugs that met this criterium, overall involving 971 cell lines with 710 CFEs (GC mutations and RACSs) and 78 CAAs.

Using these data, binary matrices were built according to Eqs. (7–9).

$$M = \left\{ m_{gc,cl} \right\} \text{ with } m_{gc,cl} \in \{0, 1\} \qquad (7)$$

$$C = \left\{ c_{cna,cl} \right\} \text{ with } c_{cna,cl} \in \{0, 1\} \qquad (8)$$

$$S = \left\{ s_{cal,cl} \right\} \text{ with } s_{cal,cl} \in \{0, 1\} \qquad (9)$$

Herein, $m_{gc,cl}$ denotes the mutation state of GC $gc$, whereas $c_{cna,cl}$ defines the copy number status of RACS $cna$, and $s_{cal,cl}$ denotes the status for CAA $cal$. These elements are 0 if the alteration is absent (wild-type) or 1 if the alteration is present (mutated/copy number-altered) in cell line $cl$. GDSC-reported drug response data of 1043 cancer cell lines to 265 anti-cancer drugs, measured as $IC_{50}$ in micro-molar concentration (μM), were log-transformed and used to generate matrix $I$.

$$I = \left\{ \left(\log_{10}(i)\right)_{d,cl} \right\} \text{ with } i_{d,cl} \in \{0, 1\} \qquad (10)$$

Herein, $d$ denotes the $d$th drug and $d \in [1, D]$. Missing $IC_{50}$ values in the data were imputed using a weighted mean of $IC_{50}$ values of the four nearest neighbours using the *R* package *VIM*, for Visualisation and Imputation of Missing Values[57] (within R version 3.6). Overall, 971 cell lines with available GC mutation, RACS, CAA and $IC_{50}$ data were used. In R, a deep neural network was developed using *TensorFlow* and the Keras Deep Learning Library *kerasR*. The models were optimised using *RMSprop* optimizer with a loss function of 'mean_squared_error' (MSE) and Rectified Linear Unit (ReLU) as the neuron activation function for all layers, except the output layer, where a linear function was applied. Fully connected layers were used. For a neuron $k$, its output $z_k$ was calculated using Eq. (11).

$$z_k = f\left(\sum_x w_{xj} \times o_x + b_k\right) \qquad (11)$$

Herein, $f$ represents the activation function and $o_x$ the output of neuron $x$ at the previous connected layer of $k$, while bias and weight are respectively represented by $b_k$ and $w_{xj}$. Therefore, the overall notation for all the neurons in a layer can be written as $z = f(w \times o + b)$. When the model is training, the weights and biases are adjusted to minimise the loss function. The resultant models were built to predict $IC_{50}$ values based on the status of GCs and RACSs, collectively referred to as cancer functional events (CFEs), and CAAs. Given a CFE and CAA pair in cell line $cl$, $\{M(:, cl), C(:, cl), S(:, cl)\}$, the model predicts $I(cl)$, which is a $D$-length vector of

$IC_{50}$. In order to determine the optimal model architecture, the hyper-parameter optimisation method called *hyperas* was employed (https://github.com/maxpumperla/hyperas). This resulted in the number of neurons in the model at the 1st layer (256 or 128), for the 2nd layer (64 or 32), for the 3rd layer (16 or 8) and a batch size of 128 or 64. The last layer in the model had 265 neurons and was linearly activated. All models were trained for 1000 epochs on 80% of pan-cancer cell lines and validated on the remaining 20% of the data (5-fold cross validation). The resultant best-performing models were stored and performance was assessed by determining the loss and accuracy as a function of the number of epochs. In addition, confusion matrices were built by evaluating the model on the test dataset using a probability threshold of 0.6 and the harmonic mean of precision and recall, $F_1$ score, was computed according to Eq. (12).

$$F_1 = \begin{cases} 0 & \text{if precision}|\text{recall} = 0 \\ \frac{2}{\text{precision}^{-1} + \text{recall}^{-1}} & \text{if precision} \cdot \text{recall} > 0 \end{cases} \quad (12)$$

**Reporting summary**. Further information on research design is available in the Nature Research Reporting Summary linked to this article.

## Data availability

Publicly available datasets used in this study, and their accession information, are: TCGA: GDC Data Portal [https://portal.gdc.cancer.gov] (unrestricted public access for the data used in this study); MSK-IMPACT: [http://cbioportal.org/msk-impact] (unrestricted public access); METABRIC: [https://www.ebi.ac.uk/ega/datasets/EGAD00010000164] (restricted access); GDSC: [https://www.ebi.ac.uk/ega/studies/EGAS00001000978] (unrestricted access). Raw data are provided in the Source Data file, Supplementary Data files and in the repository accessible via https://github.com/pascalduijf/CAAs_1. The Cancer Genome Atlas (TCGA) Level 3 Affymetrix Genome-Wide SNP6.0 Array data (version 28/01/2016), mRNA expression Illumina HiSeq RNASeq V2 log2(RSEM-normalised count + 1) data (version 13/10/2017) and clinical data were downloaded from the National Cancer Institute's Genomic Data Commons (GDC) Data Portal [https://portal.gdc.cancer.gov] for 11,019 human tumours across 31 cancer types (Supplementary Data 1). Affymetrix SNP6.0 array data from the Molecular Taxonomy of Breast Cancer International Consortium (METABRIC) trial were available for 1980 breast tumours under restricted access[22]. These data were accessed through Synapse (synapse.org) or the European Genome-phenome Archive (EGA) (https://ega-archive.org/). Memorial Sloan-Kettering-Integrated Mutation Profiling of Actionable Cancer Targets (MSK-IMPACT) SNP6 array data was available for 10,945 samples and accessed through cBioPortal [http://cbioportal.org/msk-impact][26] (Supplementary Data 2). Raw Affymetrix SNP6.0 array data for 1022 cancer cell lines were downloaded from the EGA [https://ega-archive.org/]. High-confidence cancer genes (GCs) and copy number status of recurrently copy number-altered chromosomal segments (RACSs) for both cell lines and tumours were reported by the Genomics of Drug Sensitivity in Cancer (GDSC) project and downloaded from this project's website [https://www.cancerrxgene.org/downloads][21]. Binary drug resistance/sensitivity data were also previously reported[21]. Similarly, the drug response (total of 386,293 $IC_{50}$ values) of 988 cancer cell lines towards 453 anti-cancer drugs, measured as concentration of the drug where the biological response is reduced by half ($IC_{50}$), were downloaded from the GDSC website [https://www.cancerrxgene.org/downloads], release 8.0, July 2019).

The source data underlying Figs. 1a–c, e, g, 2a–d, 3a, b, 4a–e, 5a–c and 6a, c–g are provided as a Source Data file. All the other data supporting the findings of this study are available within the article, its supplementary information or data files, via a repository at [https://github.com/pascalduijf/CAAs_1] and from the corresponding author upon reasonable request. A reporting summary for this article is available as a Supplementary Information file.

## Code availability

The source code for the analyses in this study can be accessed via https://github.com/pascalduijf/CAAs_1.

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

## Acknowledgements

We thank Dr. Nikolaus Schultz (MSKCC) for sharing a script to aid in computing segmental chromosomal copy number alterations. We thank Dr. Marianna Datseris for her constructive advice and editing the manuscript and Dr. John Kemp for supervisory support. The results of this study are in part based upon data generated by the TCGA Research Network: https://www.cancer.gov/tcga. This study makes use of data generated by the Molecular Taxonomy of Breast Cancer International Consortium. Funding for the project was provided by Cancer Research UK and the British Columbia Cancer Agency Branch. This work has been supported by the Australian Research Council Centre of Excellence for Mathematical and Statistical Frontiers (ACEMS), under grant number CE140100049 (to D.P.K.), funding from the University of Queensland Diamantina Institute and the School of Biomedical Sciences at Queensland University of Technology and a National Breast Cancer Foundation Career Development Fellowship (to P.H.G.D.).

## Author contributions

A.S., T.H.M.N., S.B.M., J.L.F. and P.H.G.D. performed the analyses. A.S., T.H.M.N., S.B.M., J.E., J.P.G., A.S.C., J.L.F. and P.H.G.D. provided scripts and/or analysis tools. A.S., T.H.M.N. and P.H.G.D. designed the study. A.S., T.H.M.N., S.B.M., H.O., K.K.K., D.P.K., L.K., E.D., J.L.F. and P.H.G.D. provided critical intellectual content for the design of the study. P.H.G.D. wrote the initial draft of the paper. H.O., D.P.K., L.K., J.L.F. and P.H.G.D. co-supervised parts of the study. P.H.G.D. conceived, designed and provided overall supervision of the study.

## Competing interests

The authors declare no competing interests.
