## [Peer Review File · Nature Communications]

Reviewers' comments:

Reviewer #1 (Remarks to the Author):

The manuscript Chromosome arm aneuploidies shape tumour evolution, cancer prognosis and precision oncology, by Shukla et al. is well written and touches on an important and novel subject. The authors identify the number of arm-level somatic copy number alterations (CAL-SCNAs) that occur across primary and metastatic tumors. They relate their numbers across tumors and the balance between gain and loss events to aspects of cancer progression, as well as the occurrence of specific events to the rate of survival of patients. Finally, they assess the value of CAL-SCNAs to predict the response of cancer cell lines to a range of anti-cancer therapies.

Several conceptual and methodological issues of the manuscript need to be addressed.

The authors compare the frequency of CAL-SCNAs across TCGA breast tumors according to their method with that obtained by Taylor et al (16) and postulate that their high correlation constitutes a technical validation of their work. However, in the Methods section they state that both methods are similar. Shouldn't they compare with other (different) approaches to count CAL-SCNAs to support the technical validation of their method?

Then, they use the comparison of CAL-SCNAs frequencies across TCGA and METABRIC breast tumors as "biological validation" of the method. First, it is hard to understand the plot in Figure S1b, since samples in both datasets are different. What do the dots in the scatterplot represent? I think a more thorough biological validation of the method should include comparison to frequencies of CAL-SCNAs computed from whole-genome sequencing data (ICGC projects or the PCAWG cohort).

The thresholds and parameters chosen in the method to compute CAL-SCNAs in tumors appear arbitrary. The rationale behind them should be explained.

The authors don't explain how they deal with whole-genome duplications, which are recurrent in tumors and which may interfere with some of their findings. For example, whole-genome duplications become more frequent as tumors progress, so the increase in the relative frequency of losses that the authors observe at more advanced clinical stages may actually occur against the background of a ploidy higher than 2, in which they may carry no biological significance. Also, the frequency of whole-genome duplications is higher in metastatic tumors (Priestley 2018, www.biorxiv.org/content/10.1101/415133v2), which may also affect their observations.

Could the relatively good results obtained by the authors in the prediction of sensitivity to anti-cancer drugs based on a model trained on CAL-SCNAs be explained by specific sensitivities generated by specific losses (following a synthetic lethal model)?

From Fig. 1B, CAL-SCNAs appear pervasive across tumors from different cancer types, including some known to be driven primarily by point mutations and relatively silent in SCNAs (such as colorectal). What is the role of CAL-SCNAs in these tumors? Do they drive tumorigenesis, or are they just a consequence of de-regulation of DNA repair mechanisms?

To support their evolutionary model, the authors could order the CAL-SCNAs observed across tumors, exploiting the ideas presented at <https://www.biorxiv.org/content/10.1101/161562v3>.

Is the increase in CAL-SCNAs observed throughout stages a consequence of the relationship between high aneuploidy and low immune infiltration described by Davoli et al (ref. 15)?

While the authors propose a model of CAL-SCNAs emergence along tumor evolution that starts with gains and follows with losses in solid tumors, the exemplary evolutionary paths highlighted from Fig. 2D all start with losses.

Reviewer #2 (Remarks to the Author):

This manuscript described chromosome arm level somatic copy number alterations in human cancer. The authors presented the overall landscape of CAL-SNCAs and several interesting observations, including that certain CAL-SNCAs strongly predict patient survival. The study is of potential significance for understanding cancer, however several questions need to be addressed.

Major comments :

1) In figure 1f, the distribution of CAL gain vs loss in different stages of cancers are statistically different judging from p values, however the actual differences among different stages are not that great (~40% vs ~32%). To propose a model (figure 1g) based on this – “both blood and solid cancers initially gain few chromosome arms, whereas only solid cancers subsequently preferentially lose chromosome arms.” may have been overinterpreting such data.

On one hand, in reference 18, it is cited that “Deletions of chromosome 3p are frequent and early events in the pathogenesis of uterine cervical carcinoma”, Also in fig2d, it seems many breast cancer starts with chromosome loss. How should readers reconcile this with model in figure 1g that chromosome arm gains often happen first?

On the other hand, the observation (Fig 1f) that late stage cancers show more chromosome loss does not strictly imply that such cancers actually evolved from early stage cancers that show slightly more chromosome arm gains. Theoretically, it could be the result that cancers of certain tissues that typically show more chromosome arm loss are also more prone to be late stages. To claim that solid tumors generally gain chromosome arms before losing chromosome arms seems inappropriate, if no further evidence is provided.

2) In Fig 4d, out of the 4 panels, only the lower right one is consistent with data provided on Table S10b. For the other three panels, there are significant difference between the figure and data in Table S10b. In addition, for many of the CAL-drug pairs, the difference in IC50 is not that great, even though statistically it is. For example, for cell lines with and without +5q, the IC50 values towards XMD8.85 are 0.98 and 1.07. To count them as one of the cases where CAL-SCNA predicts drug sensitivity is debatable. There are several other such cases in Table S10b.

Both the inconsistencies between figure and data table, as well as what is counted as change of drug sensitivity need to be addressed. Also, for both Fig4d and Table S10, the number of cell lines with and without indicated CAL-SCNA should be clearly presented. It is also important to address whether the tissue type of cancer cell lines contributes to drug sensitivity difference. For example, if leukemia cell lines commonly exhibit gain of a certain chromosome arm, and leukemia cell lines are generally more sensitized to drug X, will this lead to a conclusion that gain of that chromosome arm in general sensitize cancer cells to drug X?

3) According to Table S10b, loss of 17q significantly sensitized cells to THZ. An important question is whether on the GDSC website, a gene on 17q or a focal deletion of a segment of 17q already predicted such drug sensitivity. Without such analysis, it's hard to judge whether chromosome arm level-SNCA actually outperforms individual gene mutation/deletion and copy number changes of chromosomal segment, in terms of its ability to predict drug sensitivity.

4) One of the major findings of the GDSC project was that it could predict oncogene-associated drug sensitivities. It is hard to imagine that for EGFR inhibitor sensitivity, EGFR mutation/amplification status would perform worse than CAL-SCNA. In the manuscript it was mentioned in order to be included in the analysis of figure 4e, “at least 15% of the cell lines with IC50 data were identified as sensitive to that drug”. This criteria seems arbitrary. What kind of gene-drug interactions were excluded by such a criteria? Also for figure 4e, what kind of drug sensitivity change was considered significant? For figure 4e, the current version is very abstract. It is necessary to list the specific details of drug sensitivity calls.

5) If an aneuploid cancer has on average 4n DNA content, however for chromosome 3p the content is 2n, two questions will arise. a) For such cases, is the cancer characterized as loss of 3p,

even though cancer cells still have two copies of 3p genes? b) Using the author's method, can such cancers be discerned from cancer cases with 2n DNA content but losing one copy of 3p? Such information should be clearly provided in the manuscript.

Minor comments :

1) Although the term "blood cancer" is commonly used and easily understood, "hematological malignancies" is a more accurate term.

2) It is not clear from the manuscript whether chromosome arm loss refers to heterologous loss, homozygous loss, or both, and whether they are analyzed as one group or separate groups.

3) It has been established that p53 reduces the viability of aneuploid cancer cells. Figure 1 b shows that hematopoietic malignancies shows less CAL-SCNAs than solid tumors. A discussion should be provided here if such an observation is connected to the fact that hematopoietic malignancies have far lower p53 mutation rate than solid cancers.

Reviewer #3 (Remarks to the Author):

NCOMMS-19-12500: Chromosome arm aneuploidies shape tumor evolution, cancer prognosis and precision oncology, A. Shula et al.

This is an interesting retrospective study that seeks to elucidate the role of SCNAs that occur on the level of the chromosome arm. As the authors state, it is largely not known how such major events affect tumorigenesis. Generally, I fail to see how this directly impacts 'precision oncology' (as of now), and I would ask the authors to remove this wording from the title. Also, reading the abstract, both at first, and after reading the paper, it remains unclear why to study such larger-scale molecular aberrations in the first place, and where (and why) they occur in natural Darwinian evolution. Along these lines, I am uncertain what the study of CAL-SCNAs means as a novel aspect of tumor evolution; is there an obvious selection pressure at work? Is it negative or positive selection, or both? Next, the abstract mentioned that identification of the order of events, but it might be important to point out that this is probabilistic. Last comment regarding the abstract: 'we use machine learning' is a sadly imprecise statement. Please be more specific regarding the model you trained, how that was done (supervised, unsupervised?), and how many parameters were involved, and on what partition of the data. In the conclusions, I would like to see the authors further elaborate on the potential pitfalls of their neural net-based learning approach using large, but potentially less meaningful cell-line data, and what the next steps could be taking this to preclinical models. In this context, I think knowledge of the actual functional changes resulting from CAL-SCNAs in the somatic evolutionary process would be important. What would these events mean within the context of a complex tumor micro-environment? Could there be simpler, but potentially more powerful mathematical and statistical models that will be able to predict the impact of these molecular changes?

Specific comments:

P. 7, L. 155: what do you mean by "robust"?

P. 13, L. 300. While I see the value of 'machine learning' to reveal structure in existing large(-ish) data sets, I fail to see the wider 'clinical value'. Please elaborate and/or rephrase.

P. 16, L. 351-352. "Coherence with between data sets" is unclear and or grammatically incorrect.

Figure 1: In an effort to explain the more general origin and meaning of CA-SCNA events in somatic evolution and tumor development, it might make sense to move panel g to the front.

Figure 2: It is very hard to make sense of panel d. Could this be simplified?

Figure 3: Consider splitting this complicated figure in top two main figures, it is very busy and hard to follow as is. Could the sun bursts charts become tables?

Figure 4: Panel c is insanely complicated and only shows (to me) that someone did something with a computer.

I recommend major revisions.

We are grateful for the Reviewers' constructive comments and questions, as our addressing them has improved the quality of our manuscript. Below, we summarise the major changes in the revised manuscript (highlighted in yellow in the manuscript file) and respond to each of the Reviewers' comments.

Summary of changes

Major revisions

1. More extensive method validation
2. Clarification of methodology and justification for cut-offs
3. New: Extensive additional analyses to account for whole-genome doubling (WGD)
4. Accounted for numbers of cell lines and effect sizes in identification of CAA-drug interactions
5. New: Expanded number of drugs, from 265 (2016 dataset) to 453 drugs (2019 dataset)*
6. New: Identification of potential synthetic lethal genomic interactions that sensitize to drugs

* During the revision of our manuscript, in July 2019, an expanded drug dataset was released (see <https://www.cancerrxgene.org>). We now include these new data in our analyses so that our manuscript is as up to date as possible.

Figure Panels

Revision	Original	Change
Fig. 1f	Fig. 1g	order
Fig. 1g	Fig. 1f	order; modified: added median CAA burden per stage
Fig. 2d	Fig. 2d	modified: removed green lines
Fig. 3a, b	Fig. 3a, b	modified: added WGD univariate and multivariate Cox-ph analyses
Fig. 4a-c	Fig. 3c-e	moved
Fig. 4d, e	Fig. 3f,g	modified: added WGD univariate Cox-ph analyses
Fig. 5a, b	Fig. 3h, i	moved
Fig. 5c	n/a	new: added WGD univariate and multivariate Cox-ph analyses
Fig. 6a, b	Fig. 4a, b	moved
Fig. 6c, d	Fig. 4c, d	modified: accounted for effect sizes and numbers of cell lines and expanded the dataset from 265 to 453 drugs
Fig. 6e, f	n/a	new: comparison between performance of CFEs and CAAs

Supplementary Figure Panels

Revision	Original	Change
Fig. S1a, c, e	n/a	new: more extensive technical and biological method validation
Fig. S2a-d	n/a	new: CAA frequencies in samples with and without WGD
Fig. S3a-d	n/a	new: gains and losses as a function of CAA burden and WGD
Fig. S4a-d	Fig. 2a-d	order; modified: added median CAA burden per stage
Fig. S4e-g	n/a	new: intra-tumour gain:loss ratios per cancer type and WGD status
Fig. S5a-d	Fig. S3a-d	order
Fig. S6a-e	n/a	new: CAA burden primary/metastatic tumours with/without WGD
Fig. S7-S12	Fig. S4-9	order

Supplementary Tables

Revision	Original	Change
Table S4	Table S4	modified: added WGD univariate and multivariate Cox-ph analyses
Table S5	Table S5	modified: added WGD univariate and multivariate Cox-ph analyses
Table S10	Table S10a-d	modified: identification of CAA-drug interactions on expanded dataset (265 -> 453 drugs)
Table S11	n/a	new: identification of potential synthetic lethal and synergistic resistance pharmacogenomic interactions
Table S12	n/a	new: pharmacogenomic interactions of co-lost and co-gained focal copy number alterations on the same chromosome arm
Table S13	n/a	new: CAA and CFE model performance per drug

Point-by point response

Below, we respond the reviewers' specific comments in blue font.

Reviewers' comments:

Reviewer #1

The manuscript Chromosome arm aneuploidies shape tumour evolution, cancer prognosis and precision oncology, by Shukla et al. is well written and touches on an important and novel subject. The authors identify the number of arm-level somatic copy number alterations (CAL-SCNAs) that occur across primary and metastatic tumors. They relate their numbers across tumors and the balance between gain and loss events to aspects of cancer progression, as well as the occurrence of specific events to the rate of survival of patients. Finally, they assess the value of CAL-SCNAs to predict the response of cancer cell lines to a range of anti-cancer therapies.

Several conceptual and methodological issues of the manuscript need to be addressed.

The authors compare the frequency of CAL-SCNAs across TCGA breast tumors according to their method with that obtained by Taylor et al (16) and postulate that their high correlation constitutes a technical validation of their work. However, in the Methods section they state that both methods are similar. Shouldn't they compare with other (different) approaches to count CAL-SCNAs to support the technical validation of their method?

> We thank the reviewer for this comment and we agree. In the revised manuscript, in addition to the comparison to the Taylor *et al.* (2018) *Cancer Cell* data (Pearson $r = 0.9280$; Figure S1b), we now compare our CAL-SCNA (now simply called CAA) frequencies (determined from TCGA SNP6 array data) to:

- (2) TCGA whole genome microarray data (Figure S1c). This comparison shows a Pearson correlation of $r = 0.9106$ (Figure S1c). Since these two analyses were performed on the same biological samples, this constitutes technical validation.

Then, they use the comparison of CAL-SCNAs frequencies across TCGA and METABRIC breast tumors as "biological validation" of the method. First, it is hard to understand the plot in Figure S1b, since samples in both datasets are different. What do the dots in the scatterplot represent? I think a more thorough biological validation of the method should include comparison to frequencies of CAL-SCNAs computed from whole-genome sequencing data (ICGC projects or the PCAWG cohort).

> We agree that we could have explained this better. The data in original Figure S1b (now Figure S1d) show on the x-axis the CAL-SCNA (now called CAA) frequencies, as determined using:

- (3) The METABRIC SNP6 array dataset (Figure S1d) and on the y-axis the CAA frequencies, as determined using our method and the TCGA-BRCA SNP6 array dataset. Each dot corresponds to the frequency of the same CAA determined in the two respective datasets. We have now clarified this by including a table in Figure S1a, as well as in the Figure legend. This comparison shows a Pearson correlation of $r = 0.9688$ (Figure S1d). Since these two datasets are independent, this constitutes biological validation.

> We have now also compared our CAA frequencies to those using:

- (4) PCAWG whole-genome sequencing data. This comparison shows a Pearson correlation of $r = 0.6154$ (Figure S1a, e).

The thresholds and parameters chosen in the method to compute CAL-SCNAs in tumors appear arbitrary. The rationale behind them should be explained.

> Inevitably, the cut-offs are indeed somewhat arbitrary. We justify them as follows.

- To call chromosome arm aneuploidies, for segmental copy number changes, we chose a cut-off for absolute log copy number ratio of 0.2 to account for background noise and heterogeneity. We note that this is a well-established cut-off that is widely used in the field, for example here:
 - Sack *et al.*, 2018 *Cell* (PMID: 29576454),
 - Roy *et al.*, 2016 *Cancer Cell* (PMID: 27165745),

- Zack *et al.*, 2013 *Nat Genet* (PMID: 24071852),
- Beroukhim *et al.*, 2010 *Nature* (PMID: 20164920).
- We chose a 90% cut-off for gain/loss over the length of the chromosome arm to qualify for CAA calling, as this allows for heterogeneity and deviating focal copy number aberrations. In the literature, this cut-off varies somewhat:
 - 50%: Zack *et al.*, 2013 *Nat Genet* (PMID: 24071852),
 - 90%: Roy *et al.*, 2016 *Cancer Cell* (PMID: 27165745),
 - 98%: Beroukhim *et al.*, 2010 *Nature* (PMID: 20164920).
 Thus, our cut-off of 90% seems somewhat conservative.

We have now added this information to the manuscript (page 21).

The authors don't explain how they deal with whole-genome duplications, which are recurrent in tumors and which may interfere with some of their findings. For example, whole-genome duplications become more frequent as tumors progress, so the increase in the relative frequency of losses that the authors observe at more advanced clinical stages may actually occur against the background of a ploidy higher than 2, in which they may carry no biological significance. Also, the frequency of whole-genome duplications is higher in metastatic tumors (Priestley 2018, www.biorxiv.org/content/10.1101/415133v2), which may also affect their observations.

> This is an excellent comment, also echoed by Reviewer 2 below. We thank both reviewers, as we were interested in investigating how whole-genome doubling (WGD) might relate to our observations.

> First, as now also clarified in the Methods section (page 20), we considered SNP6 copy number segments as lost, gained or unchanged with respect to the ploidy status of the sample. Hence, chromosome arm aneuploidies (CAAs) were determined independent of WGD status. Consequently, chromosome arm loss in a sample which had undergone WGD, would be considered a loss. Therefore, this Reviewer correctly comments that losses may have occurred against a background of a ploidy higher than 2. With regards to the biological significance of this, please refer to our last response to this point (page 5 below).

> WGD status was determined using ABSOLUTE (Carter *et al.*, 2012 *Nat Biotechnol*, PMID: 22544022) (Methods section 'Whole-genome doubling analyses', page 23). To assess how WGD may affect our conclusions, we have now performed a range of additional analyses. We systematically either repeated analyses for WGD-negative (WGD-) and WGD-positive (WGD+) samples separately or we included WGD status as a covariate in our multivariate survival analyses. These new analyses are included in Figs. 3a,b, 4d,e, 5c, Suppl. Figs. S2a-d, S3a-d, S4f,g, S6a-e and Suppl. Tables S4 and S5.

In summary, we conclude that WGD+ samples typically show an increased CAA burden compared to WGD- samples. We also find that WGD status affects the moderate specificity of metastasis to liver. However, WGD has a negligible effect on other aspects of metastasis, differences between solid and haematological malignancies in terms of CAA burden and gain/loss biases, and patient survival outcome. We provide the full details below.

1. WGD and CAA burden

Repeating the analysis in Figure 1a for WGD- and WGD+ samples separately shows that CAA burden is higher in WGD+ samples than in WGD- samples (Figure S2a, b). However, the main conclusion that CAA burden is higher in solid than in haematological tumours is independent of WGD status (each $p < 0.0001$; Figure S2a, b).

2. WGD and overall chromosome arm gain:loss bias

Repeating the analyses in Figure 1c for WGD- and WGD+ samples separately also shows an overall bias towards chromosome arm loss in solid tumours and a bias towards gain in haematological tumours in both WGD- and WGD+ samples and in both paired and unpaired analyses (all p values < 0.01) (Figure S2c, d). The only exception is whole genome-doubled haematological cancers, where we did not observe a statistically significant difference, possibly due to low statistical power, as very few haematological tumours undergo WGD ($n = 13$ of 171).

Thus, the bias towards chromosome arm loss in solid cancers is independent of WGD status. The bias towards gain in haematological tumours applies to at least the vast majority (158/171 = 92%) of these malignancies that do not undergo WGD.

3. WGD and chromosome arm gain:loss ratios as a function of CAA burden

Similarly, repeating the analyses in Figure 1e for WGD- and WGD+ samples separately demonstrated that solid cancers show a significant bias towards chromosome arm gain when the CAA burden is low, but a significant bias towards chromosome arm loss when CAA burden is high, irrespective of whether they had undergone WGD (Figure S3a-d).

Interestingly, there was a slight shift in the 'turning point' from bias towards gain to bias towards loss. In WGD- samples, this turning point is between 2 and 3 CAAs/sample, whereas for whole genome-doubled samples, this point also roughly doubles, as it is between 4 and 5 CAAs/sample (Figure S3a-d).

In contrast, haematological tumours almost invariably show a bias towards gain (9 of 11 data points; the other 2 showing no bias in either direction), but, similar to observations in Figure 1e, this is rarely statistically significant for individual data points.

Overall, these observations show that our main conclusion that low CAA burden is significantly associated with bias towards gain, whereas higher CAA burden is significantly associated with loss, is independent of WGD status in solid cancers.

4. WGD and chromosome arm gain:loss ratios in clinical stages

In the revised manuscript, we analysed the chromosome arm gain (G) : loss (L) ratio distributions between the groups $G < L$, $G = L$ and $G > L$ per clinical stage for WGD- and WGD+ solid tumours separately (Figure S4f, g). This revealed that the fractions of tumours with more gains than losses ($G > L$) are highest in stage I tumours irrespective of WGD status. However, this difference is only statistically significant for WGD- tumours (each $p < 0.0015$; Figure S4f, g).

We also note that WGD- stage I tumours show a higher fraction of $G > L$ samples than WGD+ stage I tumours (Figure S4f, g), and that the stage I group is heterogeneous, consisting of tumours with a bias towards arm loss (CAA burden ≤ 2 if WGD-; CAA burden ≤ 4 if WGD+), as well as tumours with a bias towards arm gain (CAA burden ≥ 2 , ≥ 4 , respectively), as evidenced by the fact that the medians in the stage I, WGD- and the stage I, WGD+ groups are respectively 2 and 9 (Fig. S4f, g; see also point 3 above and Figs. 1e, S3a-d and S4a).

Thus, these observations enrich our understanding of the effects of WGD status and are consistent with our prior conclusion that the bias towards gain or loss is a function of the total CAA burden.

5. WGD and CAAs in metastasis

In Fig. 2a and Fig. S5, we found that CAA burden is significantly higher in metastases than in primary tumours and that this particularly applies to brain metastases. Newly added Suppl. Fig. S6a-c now shows that this is independent of WGD status.

In Fig. 2b and S5c,d, we identified individual CAAs that associate with metastasis. Newly added Fig. S6d now shows that WGD has a negligible effect on this.

In Fig. 2c, we found that several CAAs show moderate to strong specificity for metastasis specifically to liver, bone or brain. In Fig. S6e, we found that WGD status affects this specificity for liver metastases.

6. WGD and CAAs in patient prognosis

To assess if WGD is significantly associated with patient survival, we first performed univariate Cox proportional hazard analyses. As indicated in the newly added Figure 5c (left) and updated Tables S4 and S5, WGD is significantly ($p < 0.05$) associated with overall survival (OS) for 3 of 14 types of cancer and with disease-free survival (DFS) for 4 of 19 types of cancer. These observations are consistent with recent work (Bielski *et al.*, 2018, *Nat Genet*, PMID: 30013179).

Next, for these cancer types, we performed multivariate Cox proportional hazard modelling on all individual and co-occurring CAAs that we previously identified as significant (23 for OS and 21 for DFS), but now including WGD as a covariate. This invariably affected the level of significance, with the respective p values increasing or decreasing. However, most differences were small and only in 1 out of 44 cases this resulted in the significance level increasing to above 0.05, specifically $p=0.0570$ (see especially Fig. 5c (right), but also Tables S4, S5 and Fig. 3a,b, Fig. 4d,e).

Thus, we conclude that WGD has a negligible effect on whether individual CAAs or co-occurring CAA pairs predict good or poor overall or disease-free patient survival outcome.

> Finally, we agree that losses that may have occurred against a background of a ploidy higher than 2 may not carry biological significance. However, the precise effects are difficult to determine. On the one hand, a 4n genome provides a buffer against the negative effects of CAAs. For instance, the loss of essential genes on a chromosome arm that is lost probably has a lower negative impact on cell viability in a 4n background than in a 2n background. Consistently, we observed that CAA burden is higher in WGD+ tumours than in WGD- tumours (Fig. S6a, b).

On the other hand, it is well-established that changes in copy number status affect the stoichiometry of protein complexes, which will be affected irrespective of whether loss occurs in a diploid or tetraploid background (although to a different extent). This has been demonstrated in various contexts ranging from yeast and *Drosophila* to mouse and human cells (e.g., Oromendia *et al.*, 2012, *Genes Dev*, PMID: 23222101; Brennan *et al.*, 2019, *Genes Dev*, PMID: 31196865; recently reviewed by Zhu *et al.*, 2018, *Dev Cell*, PMID: 29486194). Additionally, our extensive newly added analyses focussed on the role of WGD (see above) show that WGD overall has a very limited effect. For example, it has a negligible effect on whether individual or co-occurring CAAs predict patient survival outcome (Fig. 5c, Tables S4, S5). This suggests that, at the organismal level, the biological difference between CAAs occurring in a 2n background compared to a 4n background is limited.

Could the relatively good results obtained by the authors in the prediction of sensitivity to anti-cancer drugs based on a model trained on CAL-SCNAs be explained by specific sensitivities generated by specific losses (following a synthetic lethal model)?

> This is an interesting question. To investigate this, we initially used a much broader approach that could simultaneously answer several questions, including this one. Specifically, we systematically tested whether (for each of the 28 cancer types and pan-cancer) any of the possible $\binom{788}{2} = 310,078$ co-occurring CFE/CAA pairs are significantly more frequently associated with drug sensitivity than with drug resistance, or *vice versa*, using Fisher's exact tests (page 12, lines 264-282). This could answer the following three questions, all related to drug response:

(1) Are there any potential synthetic lethal interactions involving pairs of CFEs or CAAs?

Such pairs would be associated with increased drug sensitivity. Using cut-offs of $p < 0.05$ and $FDR\% < 0.001$, we identified 1024 potential synthetic lethal interactions (Table S11).

(2) Are there any other synergistic interactions involving pairs of CFEs or CAAs?

This could similarly include interactions of pairs of CFEs/CAAs that are synergistically associated with increased resistance. Using above cut-offs, we identified 89 of such potential synergistic resistance interactions (Table S11).

(3) Can any individual CAA associated with drug response be explained by a pair of focal gains or losses (in the same direction), including synthetic lethal interactions of focal aberrations, on the same chromosome arm?

Even at a lenient cut-off of $p < 0.05$ (without any FDR cut-off), only 98 of 5171 potential synergistic interactions involved co-loss or co-gain of focal CNAs on the same chromosome arm and none of those corresponded to any of the 64 significant CAA-drug interactions that we previously identified (Table S12, Fig. 6c, Table S10).

We note that this excludes the two aforementioned CAA-drug interactions that can be explained by focal CNAs (Table S10).

From Fig. 1B, CAL-SCNAs appear pervasive across tumors from different cancer types, including some known to be driven primarily by point mutations and relatively silent in SCNAs (such as colorectal). What is the role of CAL-SCNAs in these tumors? Do they drive tumorigenesis, or are they just a consequence of de-regulation of DNA repair mechanisms?

> If these CAL-SCNAs (CAAs) drive tumorigenesis, then they would be expected to forecast poor patient outcome. In Tables S4 and S5, we have identified CAAs that significantly predict poor prognosis using multivariate survival analysis, suggesting that these CAAs drive tumorigenesis. However, we note that we identified fewer significant CAAs for cancer types typically driven primarily by point mutations, such as colorectal cancer. Thus, our data are consistent with these cancers' being primarily driven by mutations and only (perhaps secondarily) by a few CAAs.

To support their evolutionary model, the authors could order the CAL-SCNAs observed across tumors, exploiting the ideas presented at <https://www.biorxiv.org/content/10.1101/161562v3>.

> This is an excellent suggestion. Unfortunately, however, this requires the availability of whole-genome sequencing data, which are not available from the TCGA dataset. Nonetheless, we were able to utilise the evolutionary timing of CAAs that was determined in said manuscript using the ICGC/PCAWG dataset. Combining this timing with our probabilistic model, we ordered the CAAs as shown in Figure R1 below.

> We have not included this figure in our manuscript, because we are not confident about the accuracy of this integrative approach, combining observations from two different datasets. For several CAAs, the prevalence differed considerably between the TCGA dataset and the ICGC/PCAWG samples used by Gerstung *et al.*, 2017 *bioRxiv*. Most notably, the prevalence of -17q (the only early-occurring CAA)

was 31% and 80%, respectively, and $-3p$ respectively occurred in 8% and 54% of the samples. These differences may in part be caused by vastly different samples sizes of $n=1093$ and $n=97$, respectively.

Is the increase in CAL-SCNAs observed throughout stages a consequence of the relationship between high aneuploidy and low immune infiltration described by Davoli et al (ref. 15)?

> This is an intriguing question. This may well be the case, as we also observe this inverse correlation. In solid tumours, CAA burden is lowest in stage I tumours and increases with stage. Conversely, levels of infiltrating immune cells are typically highest in stage I tumours (**Figure R2a**). Irrespective of stage and type of immune cells, CAA burden and the levels of infiltrating immune cells are consistently anti-correlated (**Figure R2b**). These observations are consistent with various previous reports, including Davoli *et al.*, 2017 *Science* PMID 28104840; Taylor *et al.*, 2018 *Cancer Cell* PMID 29622463; Buccitelli *et al.*, 2017 *Genome Res* PMID 28320919.

While the authors propose a model of CAL-SCNAs emergence along tumor evolution that starts with gains and follows with losses in solid tumors, the exemplary evolutionary paths highlighted from Fig. 2D all start with losses.

> We thank this Reviewer for noting this. This was a consequence of coincidence and an error. We erroneously used $(-19q \rightarrow +1q)$ as an example and corrected this to $(+19q \rightarrow +1q)$, starting with gain. Another example that we previously included, $(-7q \rightarrow +3q \rightarrow -13q \rightarrow +8q)$, is often preceded by $7p$ gain: $(+7p \rightarrow -7q \rightarrow +3q \rightarrow -13q \rightarrow +8q)$. We corrected and added these in the revised manuscript (page 8). There are other examples that start with gain, such as $+3q \rightarrow -13q$, which we have not added.

Reviewer #2

This manuscript described chromosome arm level somatic copy number alterations in human cancer. The authors presented the overall landscape of CAL-SNCAs and several interesting observations, including that certain CAL-SNCAs strongly predict patient survival. The study is of potential significance for understanding cancer, however several questions need to be addressed.

> We thank the Reviewer for her/his interest and for recognising the potential significance of our work.

Major comments:

1) In figure 1f, the distribution of CAL gain vs loss in different stages of cancers are statistically different judging from p values, however the actual differences among different stages are not that great (~40% vs ~32%). To propose a model (figure 1g) based on this – “both blood and solid cancers initially gain few chromosome arms, whereas only solid cancers subsequently preferentially lose chromosome arms.” may have been overinterpreting such data.

> We thank the Reviewer for this comment and agree that the actual percent difference in former figure 1f (now Fig. 1g) is small. At the top of the panel in Fig. 1g, we have now added the median number of chromosome arm aneuploidies (CAAs) for each stage. This shows that the median CAA burden for stage I cancers is 4. In Fig. 1e, we found that there is a bias towards arm gain only for tumours with a CAA burden of 1 or 2. Thus, with a median CAA burden of 4, even stage I tumours are a heterogeneous group and include both tumours with a bias towards loss and tumours with a bias towards gain. This provides an explanation for the small percent difference compared to stage II-IV tumours.

> We apologise for having given the impression that we base our model (Fig. 1f) on the data in Fig. 1g. Our conclusion was based on the data in Fig. 1e. We show the data in Fig. 1g only in support of this conclusion. To clarify this, we have reversed the order of panels f and g (formerly g and f, respectively). This also addresses one of Reviewer 3's comments.

On one hand, in reference 18, it is cited that “Deletions of chromosome 3p are frequent and early events in the pathogenesis of uterine cervical carcinoma”, Also in fig2d, it seems many breast cancer starts with chromosome loss. How should readers reconcile this with model in figure 1g that chromosome arm gains often happen first?

> These are indeed interesting points. The cited work is reconcilable with our findings. The study identified 3p loss as a frequent and early event in cervical cancer using a targeted approach, focussing on several loci on 3p and 9p21, as well as the *RB* and *TP53* alleles. Since copy number status of other loci (or indeed chromosome arms) was not investigated, it is possible that one or more other chromosome arm aneuploidies (including gains) occurred even earlier than 3p loss.

It was recently shown that, in an isogenic background, 3p loss does not promote proliferation of primary human lung cells (Taylor *et al.*, 2018 *Cancer Cell* PMID 29622463). This also suggests that additional aberrations are required. These might include preceding chromosome arm gains.

> Reviewer 1 (last point) also suggested that many breast cancers seem to start with chromosome loss, based on the examples that we provided. We have now clarified this in the text (manuscript, page 8 and page 7 above). Please also refer to our above response to this point.

On the other hand, the observation (Fig 1f) that late stage cancers show more chromosome loss does not strictly imply that such cancers actually evolved from early stage cancers that show slightly more chromosome arm gains. Theoretically, it could be the result that cancers of certain tissues that typically show more chromosome arm loss are also more prone to be late stages. To claim that solid tumors generally gain chromosome arms before losing chromosome arms seems inappropriate, if no further evidence is provided.

> We thank the Reviewer for noting this and agree that this is a possibility. To investigate this, we have studied the fractions of 'G>L cancers' per clinical stage in individual solid cancer types, also comparing this to the mean CAA burden in these cancers. The heatmaps in Fig. S4e show that the G>L fractions typically decrease between stage I and II or between stage II and III (Fig. S4e, left), whereas the mean CAA burden typically increases between these stages (Fig. S4e, right). This is consistent with our general conclusion.

2) In Fig 4d, out of the 4 panels, only the lower right one is consistent with data provided on Table S10b. For the other three panels, there are significant difference between the figure and data in Table S10b. In addition, for many of the CAL-drug pairs, the difference in IC50 is not that great, even though statistically it is. For example, for cell lines with and without +5q, the IC50 values towards XMD8.85 are 0.98 and 1.07. To count them as one of the cases where CAL-SCNA predicts drug sensitivity is debatable. There are several other such cases in Table S10b.

> We thank this Reviewer for noting the inconsistencies between former Fig. 4d (now Fig. 6d) and Table S10b (now Table S10). We also agree that, despite statistical significance, some of the IC₅₀ differences were small and it is indeed debatable whether such predictions should be counted. In the revised manuscript, we have modified our approach to address this point and following other points:

1. **Increased robustness:** We have now set an additional threshold that removed all CAA-drug combinations with small effect sizes. Specifically, we determined the Glass' Δ effect size for each difference in the IC₅₀s between cell lines positive and negative for the CAA (or CFE), respectively, and now only report CAA-drug and CFE-drug interactions if their effect sizes are higher than 1 (Fig. 6c, Table S10). Indeed, this eliminated said example of 5q gain altering XMD8.85 drug response, as, although statistically significant, it was not a robust change (Fig. 6c, Table S10).
2. **Expanded drug dataset:** During the revision of our manuscript, in July 2019, an expanded drug dataset (453 instead of the previous 265 drugs) was released (see **Table R1** and <https://www.cancerrxgene.org>). We now also include these new data in our analyses. This enabled us to identify new CAA- and CFE-drug interactions (Fig. 6c, Table S10) and to have our manuscript be as up to date as possible.
3. **Reproducibility:** Our machine learning pipeline is now aligned with the one that GDSC originally published (Iorio *et al.*, 2016 *Cell* PMID: 27397505; Cokelaer, *et al.*, 2018 *Bioinformatics* PMID: 29186349), thus facilitating reproducibility.
4. **Cell lines:** We now include detailed information about the numbers of involved CAA-positive and -negative cell lines in new columns in Table S10, as well as in the bubble volcano plot in Fig. 6c (former Fig. 4d).
5. **Tissue specificity:** Our previous results were 'pan-cancer' results and therefore did not account for tissue/cancer type. We have now accounted for this, as detailed in both Fig. 6c and Table S10.

In our revision, we have now removed any inconsistencies in Tables and Figures.

Table R1. Comparison of pharmacogenomic data in previous and current manuscript		
	Previous version, July 2016 dataset*	Revised manuscript, July 2019 dataset*
Input		
Number of drugs	265	453
Number of IC ₅₀ values	224,510	386,293
Number of cell lines	1074	988
Effect size cut-off	none	1.00
Accounted for cancer type	no	yes
Output		
Number of CFE-drug interactions	252	301
Number of CAA-drug interactions	35	64
Total number of pharmacogenomic interactions	287	365
* Please refer to https://www.cancerrxgene.org .		

Both the inconsistencies between figure and data table, as well as what is counted as change of drug sensitivity need to be addressed. Also, for both Fig4d and Table S10, the number of cell lines with and without indicated CAL-SCNA should be clearly presented. It is also important to address whether the tissue type of cancer cell lines contributes to drug sensitivity difference. For example, if leukemia cell lines commonly exhibit gain of a certain chromosome arm, and leukemia cell lines are generally more sensitized to drug X, will this lead to a conclusion that gain of that chromosome arm in general sensitize cancer cells to drug X?

> We thank the Reviewer for these excellent suggestions. Please refer to our response to the previous point. That addresses this point in the following ways:

- We now use an effect size cut-off (>1), thereby eliminating all CAA-drug interactions with small IC₅₀ differences, even if the interactions were statistically significant.

- Detailed information about the numbers of involved CAA-positive and -negative cell lines are now available both in columns in Table S10 and in the bubble volcano plot in Fig. 6c (former Fig. 4d).
- Tissue/cancer type information is now also presented in both Fig. 6c and Table S10.

3) According to Table S10b, loss of 17q significantly sensitized cells to THZ. An important question is whether on the GDSC website, a gene on 17q or a focal deletion of a segment of 17q already predicted such drug sensitivity. Without such analysis, it's hard to judge whether chromosome arm level-SNCA actually outperforms individual gene mutation/deletion and copy number changes of chromosomal segment, in terms of its ability to predict drug sensitivity.

> We thank the Reviewer for this excellent comment. To investigate this, we have now used two additional approaches:

1. We added an additional column to Table S11, which lists the genomic locations of all focal copy number alterations (CNAs) that predict drug response. We then assessed if any focal CNAs overlap with CAAs that we identified, involving:
 - a. changes in the same direction (co-gain or co-loss),
 - b. response to the same drug, and
 - c. involving the same tissue/cancer type.

Accordingly, we identified two CAA-drug interactions that can be explained by focal CNAs:

- i. Both 4q loss and several focal copy number losses on 4q predict resistance to the Aurora kinase inhibitor ZM447439 in kidney renal clear cell carcinoma (KIRC), and
- ii. Both 17p loss and several 17p11-p13 focal copy number losses predict increased resistance to MCL-1 Inhibitor Molecule 1 (MIM1) in leukaemia (LAML).

In Fig. 6c, we have now highlighted these two CAAs with an asterisk. In Table S11, we have now highlighted these CAAs and corresponding focal events in red in the new column entitled: "Non-overlapping focal CNA and CAA".

2. We also systematically tested whether (for each of the 28 cancer types and pan-cancer) any of the possible $\binom{788}{2} = 310,078$ co-occurring CFE/CAA pairs are significantly more frequently associated with drug sensitivity than with drug resistance, or *vice versa*, using Fisher's exact tests (page 12, lines 276-282). This could reveal if any individual CAA associated with altered drug response can be explained by a pair of focal gains or losses (on the same chromosome arm and in the same direction as the CAA). Even at a lenient cut-off of $p < 0.05$ (without any FDR cut-off), only 98 of 5171 potential synergistic interactions involved co-loss or co-gain of focal CNAs on the same chromosome arm. We found that none of those corresponded to any of the 64 significant CAA-drug interactions that we previously identified (new Table S11, Fig. 6c, Table S10).

Thus, overall, we find that only two CAA-drug interactions can be explained by focal CNA-drug interactions (highlighted in Fig. 6c and Table S11).

4) One of the major findings of the GDSC project was that it could predict oncogene-associated drug sensitivities. It is hard to imagine that for EGFR inhibitor sensitivity, EGFR mutation/amplification status would perform worse than CAL-SCNA. In the manuscript it was mentioned in order to be included in the analysis of figure 4e, "at least 15% of the cell lines with IC50 data were identified as sensitive to that drug". This criteria seems arbitrary. What kind of gene-drug interactions were excluded by such a criteria? Also for figure 4e, what kind of drug sensitivity change was considered significant? For figure 4e, the current version is very abstract. It is necessary to list the specific details of drug sensitivity calls.

> We thank the reviewer for this comment. We agree and our data are consistent with the GDSC finding that EGFR mutation and amplification are the strongest predictors for EGFR inhibitor sensitivity. First, our results also show significant and robust associations between EGFR inhibitor sensitivity and EGFR mutations (5 times) and focal EGFR amplification (8 times, listed as "gain_cnaPANCAN124") (Table S10, Fig. S6c). Second, our newly added more in-depth analyses compare performance of a model based on CAAs to a model based on CFEs, now per drug, and this shows that while CAAs overall outperform

CFEs (30 out of 39 drugs, Fig. 6g, Table S13), CFEs still perform better for a minority of drugs (9 of 39). Indeed, the latter include EGFR inhibitors (Table S13).

> Regarding the 15% sensitive cut-off, we added the following justification to the Methods section (pages 29, lines 655-665):

"Importantly, across cell lines, the vast majority of calls were "resistant" and a minority of calls were "sensitive". Artificial balancing of the drug response data assumes equal probabilities of finding resistance and sensitive cell lines. Since the latter was not the case, here, artificial balancing of the data would render unreliable models that would over-predict the minority class. Hence, instead an established under-sampling technique was adopted without introducing synthetic examples in the data (Dal Pozzolo et al., 2015; ref 56 in the manuscript). This method indicates that for reliable prediction of drug sensitivity, for every drug at least 10% of the calls need to be sensitive. Furthermore, in order to have this minimum number of sensitive cell lines per drug so that there are sufficient examples to split between training and test sets, this worked out to be at least 15% sensitive cell lines for each drug. There were 39 drugs that met this criterium, overall involving 971 cell lines with 710 CFEs (GC mutations and RACSs) and 78 CAAs."

> We now include more in-depth analyses. Rather than only showing overall performance of the models, we have now tested CFE and CAA model performance for each of the 39 drugs using binary resistant/sensitive calls for all 971 cell lines, as previously reported (Iorio et al., 2016 Cell PMID: 27397505). Specifically, we assessed performance of both models using a quantitative metric, F₁ score, which combines precision and recall (see Methods, page 31, lines 702-705). We have now included these details for each of the 39 drugs in Fig. 6g and Table S13. In conclusion, consistent with our previous findings, CAAs perform better than CFEs, both quantitatively (30 out of 39 drugs), and qualitatively (difference in F₁ scores) (Fig. 6g, Table S13).

5) If an aneuploid cancer has on average 4n DNA content, however for chromosome 3p the content is 2n, two questions will arise. a) For such cases, is the cancer characterized as loss of 3p, even though cancer cells still have two copies of 3p genes? b) Using the author's method, can such cancers be discerned from cancer cases with 2n DNA content but losing one copy of 3p? Such information should be clearly provided in the manuscript.

> This is an excellent comment. We considered chromosome arms as lost, gained or unchanged with respect to the ploidy status of the sample. Hence, chromosome arm aneuploidies (CAAs) were determined independent of whole-genome doubling (WGD) status. Accordingly, in Reviewer's example (a), a cancer with 4n DNA content and two copies of 3p is considered to have lost 3p. Regarding (b), yes, 4n cancers with two copies of 3p can be discerned from cancer cases with 2n DNA content with loss of one copy of 3p, because the former has undergone WGD (WGD+), whereas the latter has not (WGD-). In our revised manuscript, we have clarified the above in our methodology (page 20, 21, lines 461-464) and we have now performed extensive analyses on WGD- and WGD+ samples (Figs. 3a,b, 4d,e, 5c, Suppl. Figs. S2a-d, S3a-d, S4f,g, S6a-e and Suppl. Tables S4 and S5), see also our above response to Reviewer 1, point 4 (pages 3-5 above).

Minor comments:

1) Although the term "blood cancer" is commonly used and easily understood, "hematological malignancies" is a more accurate term.

> We appreciate this suggestion and have substituted the term "blood cancer" throughout the manuscript.

2) It is not clear from the manuscript whether chromosome arm loss refers to heterologous loss, homozygous loss, or both, and whether they are analyzed as one group or separate groups.

> We have analysed these as one group and have now clarified this in the Methods of our manuscript (page 21, lines 479-480).

3) It has been established that p53 reduces the viability of aneuploid cancer cells. Figure 1 b shows

that hematopoietic malignancies shows less CAL-SCNAs than solid tumors. A discussion should be provided here if such an observation is connected to the fact that hematopoietic malignancies have far lower p53 mutation rate than solid cancers.

> We thank the Reviewer for this suggestion. This nicely adds to the other points about p53 that we had already discussed. We have now added this point on page 14 (lines 322-323).

Reviewer #3

NCOMMS-19-12500: Chromosome arm aneuploidies shape tumor evolution, cancer prognosis and precision oncology, A. Shula et al.

This is an interesting retrospective study that seeks to elucidate the role of SCNAs that occur on the level of the chromosome arm. As the authors state, it is largely not known how such major events affect tumorigenesis.

Generally, I fail to see how this directly impacts 'precision oncology' (as of now), and I would ask the authors to remove this wording from the title.

> We agree that our work does not directly impact 'precision oncology' (see also below) and believe that the term 'drug response' is more accurate. Hence, we have changed the title accordingly.

Also, reading the abstract, both at first, and after reading the paper, it remains unclear why to study such larger-scale molecular aberrations in the first place, and where (and why) they occur in natural Darwinian evolution. Along these lines, I am uncertain what the study of CAL-SCNAs means as a novel aspect of tumor evolution; is there an obvious selection pressure at work? Is it negative or positive selection, or both?

> We studied chromosome arm aneuploidies (CAA), because they are, in relative terms, much more prevalent in tumours than focal CNAs. This has long been recognized. For example, in 2010, Beroukhim *et al.* (*Nature*, PMID: 20164920) reported: "Arm-level SCNAs occur approximately 30 times more frequently than would be expected by the inverse-length distribution associated with focal SCNAs (Fig. 1a). This observation is seen across all cancer types (Supplementary Fig. 2), and applies to both copy gains and losses (data not shown). As a result, in a typical cancer sample, 25% of the genome is affected by arm-level SCNAs and 10% by focal SCNAs, with 2% overlap." In addition, CAAs have a larger impact on gene expression than focal SCNAs. Despite this long-standing knowledge, it has thus far remained largely unknown how CAAs might affect tumour development, patient prognosis and drug response. Hence, we focus on these aspects in this manuscript. We have now included this justification in the Introduction (page 4, lines 77-86).

> Although there are large differences between cancer types (Fig. 1b, Fig. S5a, b), we observed that CAA burden typically increases with clinical stage (Fig. 1g, Fig. S6a-g) and following metastasis (Fig. 2a, Fig. S5a-d, S6a-e). This suggests that CAAs are progressively acquired during tumour development. However, we cannot rule out the existence of one-off events that result in simultaneous acquisition of many CAAs, similar to phenomena like chromothripsis and whole-genome doubling (WGD). Our observation that WGD+ samples show significantly higher CAA burden than WGD- samples (Fig. S2a, b) suggests that WGD predisposes to the acquisition of CAAs. It is plausible that a 4n genome provides a buffer against the negative effects of CAAs. For instance, the loss of essential genes on a chromosome arm that is lost probably has a lower negative impact on cell viability in a 4n background than in a 2n background.

> Several of our observations strongly suggest the existence of both positive and negative selection. First, while the overall CAA burden is increased in metastatic samples compared to primary samples (Fig. 2a, Fig. S5a-d, S6a-e), the frequencies of individual CAAs increase remarkably more so than others, suggesting positive selection (Fig. 2b, c, Fig. S5c, d, S6d, e). In contrast, the frequencies of some CAAs are reduced in metastases compared to type-matched primary tumours, suggesting negative selection (Fig. S5c, d, S6d, e).

Second, beyond individual CAAs, our probabilistic co-occurrence analyses show that some CAA pairs occur significantly more frequently than expected by chance ('positive'), whereas others co-occur significantly less frequently ('negative') (Fig. 4a-c, Table S6).

Third, our systematic multivariate analyses of patient survival outcome identify both individual and co-occurring CAAs, which predict poor and good overall or disease-free survival outcome (Fig. 3a, b, Fig. 4d, e, Tables S4, S5, S7). This, too, suggests that there are both positive and negative selection mechanisms at work.

It is important to note, however, that there are considerable differences between cancer types. For example, we did not identify any particular 'universal pan-cancer' CAA linked to metastasis (Fig. 2b, Fig. S5c). This indicates that cell type or tissue type affects whether and the extent to which positive or negative selection occurs. On the other hand, there are some CAAs that appear frequent among many different cancer types, such as 8q gain, which is associated with *MYC* amplification (Beroukhi *et al.*, 2010 *Nature* PMID: 20164920).

Next, the abstract mentioned that identification of the order of events, but it might be important to point out that this is probabilistic.

> We agree with this point and have added the word 'probabilistic' to the abstract to clarify this.

Last comment regarding the abstract: 'we use machine learning' is a sadly imprecise statement. Please be more specific regarding the model you trained, how that was done (supervised, unsupervised?), and how many parameters were involved, and on what partition of the data.

> We appreciate this suggestion and have now provided these details in the abstract: "*Supervised machine learning, specifically, elastic net regression using 5-fold cross-validation, 988 cell lines, 788 genomic features and 386,293 IC50 values of 453 drugs, identifies 31 CAAs that robustly alter response to 56 chemotherapeutic drugs across cell lines representing 17 cancer types*" (page 2).

In the conclusions, I would like to see the authors further elaborate on the potential pitfalls of their neural net-based learning approach using large, but potentially less meaningful cell-line data, and what the next steps could be taking this to preclinical models. In this context, I think knowledge of the actual functional changes resulting from CAL-SCNAs in the somatic evolutionary process would be important. What would these events mean within the context of a complex tumor micro-environment? Could there be simpler, but potentially more powerful mathematical and statistical models that will be able to predict the impact of these molecular changes?

> We have now addressed these points in the Discussion (pages 17-18, lines 383-415):

"Our model was exclusively based on pharmacogenomic features. Thus, we anticipate that inclusion of transcriptomic or proteomic parameters could further improve performance. In addition, a potential problem in machine learning involves the presence of confounding factors, known or unknown, which can affect model performance⁴². Such factors could include cancer subtypes, which our models did not account for. Thus, inclusion of subtype and application of confounder control methods could improve model performance⁴³.

The functional consequences of CAAs are not always easily understood. Two of the identified 64 CAA-drug interactions could be explained by focal CNAs, suggesting involvement of HNRNPD and TP53 loss (see above). Other CAA-drug interactions could not be explained by co-occurring focal CNAs on the same chromosome arm, even if distant from each other. Thus, more complex interactions exist, potentially synthetic lethal events involving three or more loci. While in vitro studies are required to understand the full consequences of individual CAAs¹⁷, our work does provide some clues. For example, 17p loss increases resistance to 7 different drugs in leukaemia (LAML) and 5 of these target cell cycle/mitotic regulators (KIF11, CDK2/7/9, WEE1, PLK1, microtubules) (Supplementary Table 10). This links 17p loss to resistance to cell cycle inhibitors. This may well involve a complex interaction with TP53, as it is located at 17p13.1, while TP53 loss (or mutation) alone is not predictive (Supplementary Table 10).

We also highlight that context matters. This applies to the broad genomic context, as evidenced by our analyses involving individual CAAs, which are akin to a large number of focal CNAs, and co-occurrence analyses, including in the context of patient survival and potential synthetic lethal interactions. There are also vast cancer type-specific differences. Additionally, the tumour microenvironment is complex, involving clonal and sub-clonal

aberrations, as well as other cell types, including tumour-infiltrating lymphocytes, whose abundance inversely correlates with aneuploidy^{16,17}. In this light, our observation that CAAs strongly predict drug response may lay a foundation for pre-clinical studies, involving validation in mouse xenograft, patient-derived cancer organoid (PDO) or xenograft (PDX) models^{44,45}. This will be critical, because even though cell lines typically well represent genetic and genomic somatic alterations found in tumours²¹, PDX models in particular much better mimic the complexities that exist in the tumour microenvironment⁴⁵. Taken together, our findings can be a starting point for pre-clinical studies and hence have the potential to ultimately advance precision oncology."

Specific comments:

P. 7, L. 155: what do you mean by "robust"?

> This is now on page 8, line 186. These analyses are 'robust', because they are not only multivariate – as opposed to univariate – but also account for many co-variates, including cancer type, clinical stage, age, all univariately significant CAAs, as well as now in our revised manuscript, WGD status.

P. 13, L. 300. While I see the value of 'machine learning' to reveal structure in existing large(-ish) data sets, I fail to see the wider 'clinical value'. Please elaborate and/or rephrase.

> This is now on page 16, line 371. We have removed the term 'clinical value' here, as we agree that there is no immediate 'clinical value'. Rather, our findings could provide a basis for pre-clinical studies, such as mouse xenograft and patient-derived xenograft (PDX) models that could ultimately result in clinical value for diagnostics and/or therapeutics. We have now discussed this in the Discussion (page 18, lines 405-415).

P. 16, L. 351-352. "Coherence with between data sets" is unclear and or grammatically incorrect.

> We thank the Reviewer for identifying this error. We have corrected this to read: "coherence between datasets" (page 20, line 452).

Figure 1: In an effort to explain the more general origin and meaning of CA-SCNA events in somatic evolution and tumor development, it might make sense to move panel g to the front.

> We thank the Reviewer for this suggestion. We have now swapped the order of panels f and g. Hence, original panel g is moved in front of original panel f. We agree that this makes more sense for two reasons.

- First, panel f in the revision is based directly on the preceding data in panels c, d and e.
- Second, this avoids that we incorrectly give the reader (including Reviewer 2; see her/his major comment 1 above) the impression that the model (now in panel f) is based on the data in revised panel g. As we detail above (Reviewer 2, major comment 1), the data in revised panel g are instead presented to support the model.

Figure 2: It is very hard to make sense of panel d. Could this be simplified?

> This original panel had red- and green-coloured edges to indicate direction (red to the right and green to the left). Strictly speaking, this is not necessary, because the red and green lines are mirror images of each other. We have now removed all green lines. This retains all critical information, while simplifying this panel.

Figure 3: Consider splitting this complicated figure in top two main figures, it is very busy and hard to follow as is. Could the sun bursts charts become tables?

> Thank you for these suggestions. We have now split former Figure 3 as follows:

Former	Current
Fig. 3a, b	Fig. 3a, b
Fig. 3c-g	Fig. 4a-e
Fig. 3h, i	Fig. 5a, b

> The data in the sunburst charts (now Fig. 5a, b) are already provided in table format (Supplementary Tables S4, S5 and S7). These three tables are too large to be included as main tables. The sunburst graphs

provide the reader with a visual summary of the data in these extensive tables. Therefore, instead of adding another table, we opted to retain the sunburst charts. We have now also clarified the information in these panels by adding the numbers of CAAs to the Figure, rather than as a description in the legend. Retaining Fig. 5a and b also enabled us to highlight the role of WGD in the newly added Fig. 5c, a major point raised by both Reviewers #1 and #2.

Figure 4: Panel c is insanely complicated and only shows (to me) that someone did something with a computer.

> To improve clarity of this panel (now Fig. 6c), we reduced the number of labels.

I recommend major revisions.

REVIEWERS' COMMENTS:

Reviewer #1 (Remarks to the Author):

The authors have carried out an outstanding work answering mine and other reviewers' comments. I think as a result the manuscript has improved significantly. I have only one remaining comment. The authors' remark made in the Abstract that "Whole-genome doubling has a negligible effect." is not founded on their results. Actually, as the authors themselves specify in the Results section: "...WGD has a negligible effect on the patient survival prognostic power of CAAs." Therefore, the incomplete statement in the Abstract is misleading and should be amended.

Reviewer #2 (Remarks to the Author):

The additional analysis and clarifications in the current version adequately addressed my initial comments. The revised manuscript will bring important insight into cancer etiology and drug response.

Reviewer #3 (Remarks to the Author):

the authors have gone through lengths answering the referee's points in detail, and did a great job.

REVIEWERS' COMMENTS:

Reviewer #1 (Remarks to the Author):

The authors have carried out an outstanding work answering mine and other reviewers' comments. I think as a result the manuscript has improved significantly. I have only one remaining comment. The authors' remark made in the Abstract that "Whole-genome doubling has a negligible effect." is not founded on their results. Actually, as the authors themselves specify in the Results section: "...WGD has a negligible effect on the patient survival prognostic power of CAAs." Therefore, the incomplete statement in the Abstract is misleading and should be amended.

> We thank the Reviewer for her/his constructive comments. We have revised the Abstract to reflect the Reviewer's remaining comment.

Reviewer #2 (Remarks to the Author):

The additional analysis and clarifications in the current version adequately addressed my initial comments. The revised manuscript will bring important insight into cancer etiology and drug response.

> We thank the Reviewer for her/his constructive comments and for valuing our work.

Reviewer #3 (Remarks to the Author):

the authors have gone through lengths answering the referee's points in detail, and did a great job.

> We thank the Reviewer for her/his constructive comments.